# LUVE : Latent-Cascaded Ultra-High-Resolution Video Generation with Dual Frequency Experts

Chen Zhao [* 1 2]   Jiawei Chen [* 1]   Hongyu Li [2]   Zhuoliang Kang [2]   Shilin Lu [3]   Xiaoming Wei [2]   Kai Zhang [1]
Jian Yang [1]   Ying Tai [† 1]

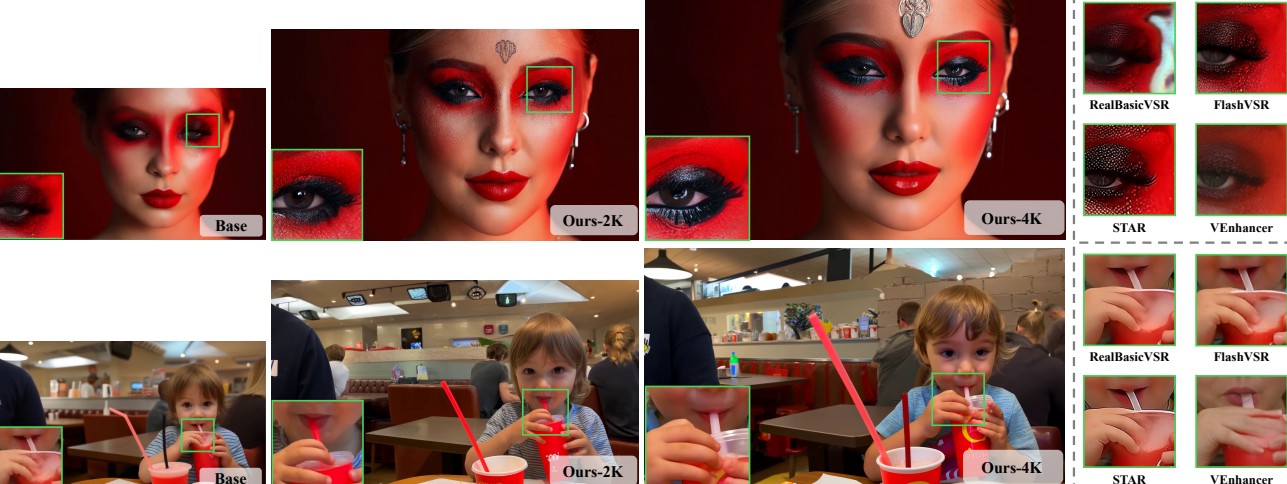

*Figure 1.* The base corresponds to the pretrained T2V model used in the first stage of our framework (Wan et al., 2025). As shown, compared with existing VSR methods, our model not only produces videos that are noticeably sharper and richer in fine details, but more importantly, it significantly enhances semantic consistency and plausibility. This demonstrates that ***UHR generation goes beyond merely enhancing visual sharpness—it fundamentally advances semantic coherence and content fidelity***. (Zoom-in for best view)

## Abstract

Recent advances in video diffusion models have significantly improved visual quality, yet ultra-high-resolution (UHR) video generation remains a formidable challenge due to the compounded difficulties of motion modeling, semantic planning, and detail synthesis. To address these limitations, we propose **LUVE**, a **L**atent-cascaded **U**HR **V**ideo generation framework built upon dual frequency **E**xperts. LUVE employs a three-stage architecture comprising low-resolution motion generation for motion-consistent latent synthesis, video latent upsampling that performs resolution upsampling directly in the latent space to mitigate memory and computational overhead, and high-resolution content refinement that integrates low-frequency and high-frequency experts to jointly enhance semantic coherence and fine-grained detail generation. Extensive experiments demonstrate that our LUVE achieves superior photorealism and content fidelity in UHR video generation, and comprehensive ablation studies further validate the effectiveness of each component. The project is available at https://github.io/LUVE/.

## 1. Introduction

Video generation has achieved considerable progress, demonstrating wide-ranging application potential in domains such as virtual reality, digital humans, and artistic creation (Kong et al., 2024; HaCohen et al., 2024; Hong et al., 2022; Yang et al., 2024). With the advancement of display technologies and the surging consumer demand for high-definition content, developing models capable of generating ultra-high-resolution (UHR) videos has become a pressing research imperative (Xue et al., 2025; Ren et al., 2025; Zhao et al., 2025b;a; Qiu et al., 2025). However, most

*Equal contribution [†]Corresponding author. [1]Nanjing University [2]Meituan [3]Nanyang Technological University. This work was done while Chen Zhao is an intern at Meituan. The project was led by Zhuoliang Kang and Hongyu Li.

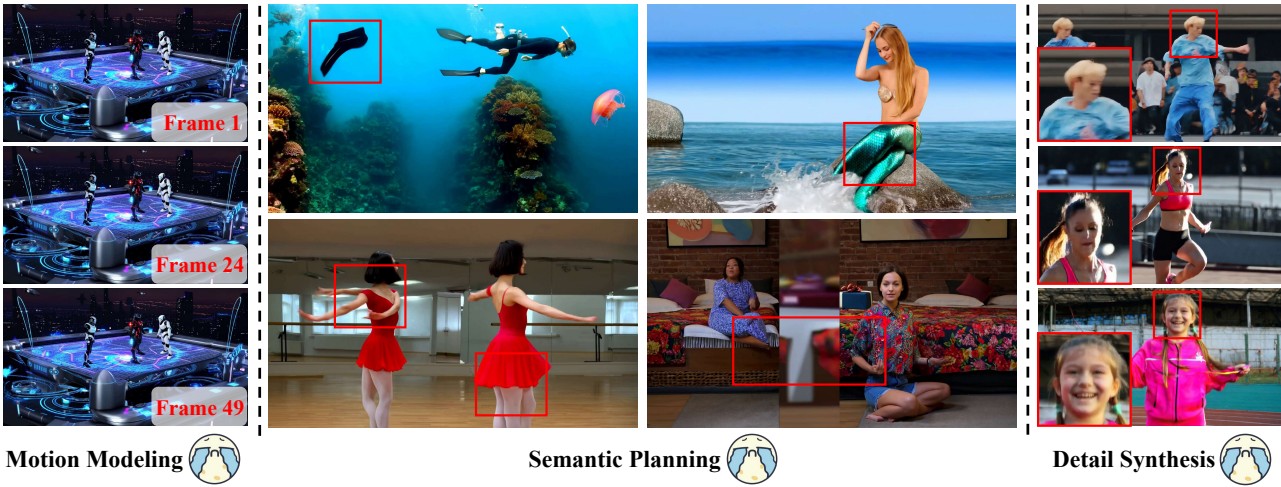

**Motion Modeling**  **Semantic Planning**  **Detail Synthesis**

*Figure 2.* Scaling T2V models to UHR scenarios introduces several challenges. In motion modeling, models tend to produce static outputs, failing to capture coherent temporal dynamics. In semantic planning, both global and local repetitions emerge, reflecting insufficient semantic understanding. Finally, in detail synthesis, the generated frames often suffer from motion blur and texture degradation.

existing models exhibit significant quality degradation when scaled to the task of UHR video generation. This limitation poses a substantial barrier to real-world applications demanding fine-grained detail and high visual fidelity.

Existing efforts to overcome this challenge can be broadly divided into two paradigms: training-free approaches (Qiu et al., 2025; Ye et al., 2025) and video super-resolution (VSR)-based models (Xie et al., 2025; He et al., 2024; Zhuang et al., 2025). Training-free methods aim to synthesize UHR videos by adapting network architectures or refining inference strategies. However, these approaches often yield over-smoothed textures and unrealistic high-frequency details, as they ultimately rely on pre-trained text-to-video (T2V) diffusion models that have never been exposed to UHR data, and therefore *lack the inherent generative capacity to reproduce the authentic UHR video*. In contrast, VSR-based approaches adopt a two-stage pipeline: first generating low-resolution videos using pre-trained T2V models, followed by spatial upscaling via specialized VSR models. Although this paradigm enhances visual clarity, such improvements are restricted to low-level textures and lack the capacity to generate meaningful semantic or structural details, as shown in Figure 1. *Consequently, the results often exhibit pseudo-high-resolution characteristics, appearing sharper yet lacking genuine realism and content richness.*

Recently, UltraVideo (Xue et al., 2025) introduced a high-quality UHR T2V dataset, enabling the training of models capable of native UHR video generation. Nevertheless, directly training UHR video generation model remains challenging due to three closely related factors: *motion modeling, semantic planning, and detail synthesis*. As illustrated in Figure 2, first, motion modeling becomes increasingly

difficult at high resolutions, where the limitations of temporal modeling in video diffusion models are amplified, often resulting in partially or entirely static outputs, particularly in complex dynamic scenes. Second, semantic planning is impeded by the expanded spatial dimensions, which can produce spatial repetition and inconsistencies. Finally, fine-grained detail synthesis represents a critical bottleneck, as high-resolution generation frequently suffers from motion blur, texture degradation, and insufficient high-frequency information. *Consequently, comprehensively enhancing the capabilities of UHR video generation remains both a significant challenge and a problem of considerable importance.*

To address these challenges, we propose LUVE, a latent-cascaded UHR video generation framework based on dual frequency experts. This framework is meticulously designed to achieve high-quality UHR video generation and is structured into three collaborative stages: low-resolution motion generation (LMG), video latent upsampling (VLU), and high-resolution content refinement (HCR). Specifically, the LMG focuses on generating motion-consistent low-resolution video latents, providing robust motion priors for high-resolution synthesis. Following this, the VLU upsamples video latents directly within the latent space through our meticulously designed video latent upsampler, avoiding the substantial memory and computation overhead of VAE codecs. Finally, the HCR integrates the proposed low- and high-frequency experts, which respectively enhance semantic coherence and fine-grained detail synthesis, yielding photorealistic and detail-rich UHR video generation.

In summary, the key contributions of our paper are summarized as follows:

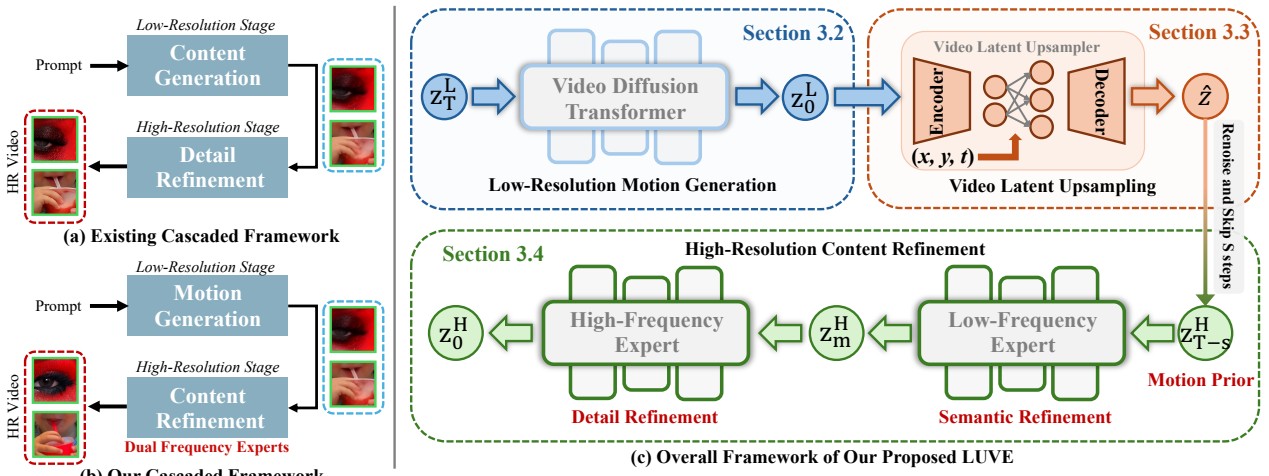

*Figure 3.* Overview of the LUVE framework. (a) and (b) illustrate the core distinction between existing cascaded high-resolution video generation architectures and our proposed paradigm. While previous methods focus on high-resolution detail refinement, our approach prioritizes high-resolution content and semantic fidelity. (c) Our LUVE, which consists of three collaborative stages: low-resolution motion generation (LMG), video latent upsampling (VLU), and high-resolution content refinement (HCR).

- We propose LUVE, a novel framework for UHR video generation, featuring a three-stage cascaded architecture that integrates LMG, VLU, and HCR to produce high-quality and detail-rich UHR videos.

- We introduce a meticulously designed video latent upsampler capable of performing arbitrary-resolution upsampling directly on video latents.

- We design dual-frequency experts, where the low-frequency expert focuses on enhancing global semantic coherence, and the high-frequency expert synthesizes fine-grained and realistic textures.

**Conflict of Interest Disclosure.** The authors declare that they have no relevant financial or non-financial conflicts of interest to disclose.

## 2. Related Works

### 2.1. Video Diffusion Models

Recent advances in video generation have achieved remarkable progress, enabling the synthesis of high-fidelity and temporally coherent videos directly from textual prompts (Yang et al., 2024; Kong et al., 2024; Lin et al., 2024; Zheng et al., 2024). Early methods extend text-to-image diffusion models by introducing temporal modules to capture frame dynamics, yet often fail to model holistic spatiotemporal dependencies (Ho et al., 2022; Guo et al., 2023; Chen et al., 2023a; Blattmann et al., 2023). With the emergence of the diffusion transformer (DiT) (Peebles & Xie, 2023), transformer-based architectures have become

the dominant paradigm, jointly modeling spatial and temporal correlations through full or interleaved attention mechanisms (Yang et al., 2024; Kong et al., 2024; HaCohen et al., 2024; Wan et al., 2025). Modern text-to-video (T2V) models typically adopt the framework consisting of a 3D VAE for spatiotemporal compression and a DiT for latent-space denoising. To balance quality and efficiency, most models employ strong latent compression while performing block-wise denoising in the latent domain (HaCohen et al., 2024). Building on this foundation, recent works, including CogVideoX (Yang et al., 2024), HunyuanVideo (Kong et al., 2024), and Wan (Wan et al., 2025), further scale up model size and data, demonstrating impressive video quality and temporal consistency at unprecedented levels. In this paper, we aim to enhance the generative capability of pretrained T2V models in UHR video scenarios.

### 2.2. Ultra-High-Resolution Visual Generation

Ultra-high-resolution (UHR) visual generation remains a fundamental challenge in visual synthesis, hindered by immense computational demands, limited high-quality data, and the scalability constraints of current models. Existing research primarily follows three paradigms: training-free approaches, fine-tuning strategies, and super-resolution frameworks. Training-free methods extend pre-trained diffusion models to higher resolutions without retraining by modifying denoising processes or attention structures, achieving computational efficiency but often producing over-smoothed textures and unrealistic high-frequency details (He et al., 2023; Du et al., 2024; Liu et al., 2024; Zhang et al., 2023; Zhao et al., 2024a; Qiu et al., 2025; Ye et al., 2025). Fine-tuning strategies adapt low-resolution generative models

on high-resolution datasets, effectively enhancing fidelity while preserving generative priors (Cheng et al., 2024; Ren et al., 2024; Chen et al., 2025a; Guo et al., 2025; Xue et al., 2025). Super-resolution-based methods employ a two-stage pipeline—low-resolution generation followed by spatial upscaling via dedicated SR or VSR networks—to recover finer details (Xie et al., 2025; He et al., 2024; Zhuang et al., 2025; Team et al., 2025; Zhang et al., 2025e; Gao et al., 2025b). However, these frameworks mainly enhance perceptual sharpness without introducing new semantic or structural content, resulting in pseudo-UHR outputs that appear sharper but lack authentic realism and richness.

## 3. Methodology

### 3.1. Overall Framework

The framework of our proposed LUVE is shown in Figure 3, which integrates three collaborative stages: low-resolution motion generation (LMG), video latent upsampling (VLU), and high-resolution content refinement (HCR). In the first stage, LMG leverages pretrained T2V model to generate motion-consistent low-resolution video latents, establishing reliable motion priors for high-resolution synthesis. VLU then performs direct upsampling within the latent space via our proposed video latent upsampler, effectively eliminating the considerable memory and computational burden of conventional VAE codecs. Finally, HCR employs our dual frequency experts, where the low-frequency expert enhances semantic coherence while the high-frequency expert refines textures and details.

**Core Novelty.** Existing cascaded high-resolution video generation models—originating from both academia (e.g., FlashVideo (Zhang et al., 2025d), LaVie (Wang et al., 2024)) and industry (e.g., Waver (Zhang et al., 2025e), Seedance(Gao et al., 2025b), Longcat-Video(Team et al., 2025))—consistently adhere to the paradigm depicted in Figure 3(a). Within this framework, the low-resolution stage is designated for foundational content synthesis, whereas the high-resolution (HR) stage is restricted to detail refinement. *This design imposes a critical bottleneck: the HR stage often functions merely as a perceptual enhancer rather than a content completer, failing to rectify semantic inaccuracies or synthesize rich content.* In contrast, our framework, illustrated in Figure 3(b). Rather than merely refining details, our high-resolution stage is specifically engineered to bolster semantic fidelity and richness within UHR scenarios, elevating the capabilities of UHR video generation.

### 3.2. Low-Resolution Motion Generation

When scaling video diffusion models to ultra-high resolutions, we observe a severe degradation in temporal dynamics, with videos in complex scenes appearing almost static

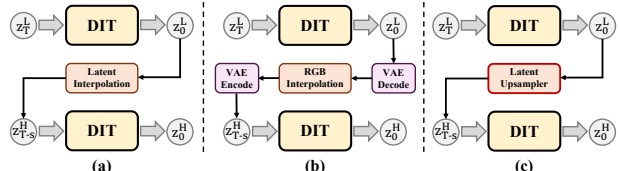

*Figure 4.* Framework Comparison. (a) Existing latent interpolation framework. (b) Existing RGB interpolation framework. (c) Our framework based on our video latent upsampler (VLUer)).

or exhibiting unrealistically small motion. This issue primarily arises because motion modeling becomes increasingly difficult at high resolutions, where the intrinsic limitations of temporal modeling in video models are further amplified. Prior studies have demonstrated that motion dynamics can be more effectively learned at lower resolutions (Zhang et al., 2025e; Gao et al., 2025b; Team et al., 2025), where models are less constrained by spatial redundancy and computational complexity. To this end, rather than directly generating high-resolution videos, we first synthesize low-resolution video latents that serve as robust motion priors for subsequent UHR synthesis. For this stage, we adopt flow matching-based video foundation model Wan2.1 as the backbone (Wan et al., 2025), which consists of a 3D VAE for spatiotemporal compression (Esser et al., 2021; Kingma, 2013), a T5-based text encoder for prompt conditioning (Raffel et al., 2020), and a transformer-based latent diffusion model for generative denoising.

### 3.3. Video Latent Upsampling

**Motivation.** As illustrated in Figure 4, existing UHR generation frameworks can be broadly categorized into latent interpolation and RGB interpolation paradigms. However, latent interpolation often results in feature distortion and manifold deviation (Ye et al., 2025), while RGB interpolation introduces blurring artifacts and incurs substantial memory and inference overhead due to VAE codecs (Qiu et al., 2025). Inspired by LSRNA (Jeong et al., 2025), we aim to learn a lightweight, trainable mapping that directly projects low-resolution video latents onto the high-resolution latent manifold. This design not only mitigates the manifold deviation inherent in traditional latent interpolation but also eliminates the additional computational and memory costs associated with VAE codecs.

**Architecture.** To achieve this goal, we design a lightweight video latent upsampler (VLUer) based on implicit neural representation (INR) architecture (Chen et al., 2021; 2022), which enables flexible and continuous upsampling to arbitrary resolutions. Our VLUer primarily consists of three components: an encoder, a video INR upsampler, and a decoder. The encoder and decoder are implemented as lightweight networks built upon temporal mutual self-attention (Liang et al., 2024), which provides low

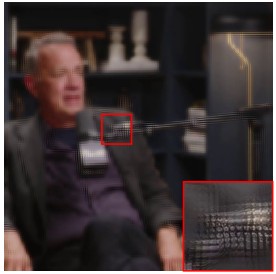 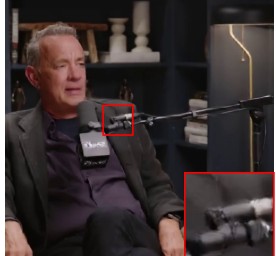

**Latent interpolation**    **w/o Decoder**

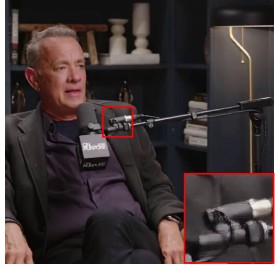 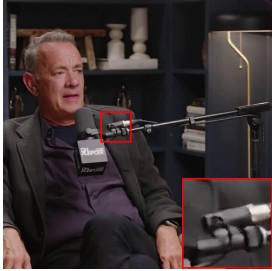

**w/o $\mathcal{L}_{pixel}$**    **Full VLUer**

*Figure 5.* Visual analysis of the key components in VLUer. These results demonstrate that our decoder effectively alleviates blurriness, while our $\mathcal{L}_{\text{pixel}}$ successfully mitigates blocky artifacts.

computational complexity while maintaining strong temporal modeling capability. Formally, the encoder takes the low-resolution latent representation $z_0^{\text{L}} \in \mathbb{R}^{t \times h \times w \times C}$ as input and extracts a feature map $F \in \mathbb{R}^{t \times h \times w \times C'}$. The extracted $F$ is then processed by the video INR upsampler to perform upsampling within the latent space. Subsequently, the decoder further learns spatio-temporal representations in the high-resolution latent domain and reconstructs the corresponding high-resolution latent representation $\hat{z} \in \mathbb{R}^{t \times H \times W \times C}$, which can be formulated as:

$$\hat{z}(x, y, t) = Decoder(U(F, Q(x, y, t))), \quad (1)$$

where $Q$ denotes the 3D coordinate, $U$ represents the video INR upsampler, and the decoder aims to further model temporal dependencies within the high-resolution latent space. As shown in Figure 5, the reconstructed video without the decoder exhibits noticeable blurriness.

**Training Target.** We initially adopt only the L1 loss between the super-resolved latent $z_{sr}$ and the high-resolution latent $z_{hr}$ as the training objective, which can be formulated as $\mathcal{L}_{latent} = \mathcal{L}_1(z_{sr}, z_{hr})$. This enables VLUer to roughly learn the latent-space upsampling mapping. However, the decoded videos still exhibit noticeable blocky artifacts. To mitigate this issue, we further incorporate the L1 loss between the decoded super-resolved video $x_{sr}$ and the high-resolution video $x_{hr}$, providing pixel-level supervision to improve reconstruction fidelity. Moreover, to enhance temporal coherence, we introduce a frame-difference loss that constrains inter-frame variations and effectively reduces

temporal flickering. The additional pixel-space loss introduced in our VLUer can be formulated as follows.

$$\mathcal{L}_{pixel} = \mathcal{L}_1(x_{sr}, x_{hr}) + \mathcal{L}_{frame}(x_{sr}, x_{hr}), \quad (2)$$

$$\mathcal{L}_{frame}(x_{sr}, x_{hr}) = \frac{1}{n-1} \sum_{t=2}^{n} \left\| \Delta x_{sr}^{(t)} - \Delta x_{hr}^{(t)} \right\|_1, \quad (3)$$

where $\Delta x_{sr}^{(t)} = x_{sr}^{(t)} - x_{sr}^{(t-1)}$ denotes the difference between two consecutive frames. As shown in Figure 5, the decoded video without $\mathcal{L}_{\text{pixel}}$ exhibits noticeable blocky artifacts.

### 3.4. High-Resolution Content Refinement

**Motivation.** Existing studies have demonstrated that the global semantic structure (low-frequency components) is reconstructed during the high-noise stage, whereas fine-grained details (high-frequency components) are progressively synthesized in the later low-noise stage (Yi et al., 2024; Zhang et al., 2024c). To investigate the frequency dynamics of video diffusion models during denoising, we perform Power Spectral Density (PSD) analysis on Wan2.1. As shown in Figure 6, the model shows a clear frequency behavior: primarily capturing low-frequency components in the high-noise stage, then gradually shifting focus toward high-frequency details in the low-noise stage. However, scaling to UHR substantially expands spatial information, resulting in low-frequency semantic structural inconsistencies and high-frequency detail degradation. The insight motivates us to design two specialized experts that operate at different denoising phases: the low-frequency expert, which enhances semantic consistency during the high-noise stage, and the high-frequency expert, which refines fine-grained details in the low-noise stage.

**Low-Frequency Expert.** Based on this motivation, we introduce the low-frequency expert (LFE), which is trained during the high-noise stage ($t \in [t_{\text{switch}}, 1]$) to enhance global semantic consistency. For parameter-efficient adaptation, we implement the LFE using low-rank adaptation (LoRA) (Hu et al., 2022). To ensure the expert explicitly focuses on the low frequency band, we apply a low-pass filter to the input features $\mathbf{x}$ before they are processed by the LoRA. It is noteworthy that the attention module within DiT blocks is primarily responsible for capturing global information. Therefore, as illustrated in Figure 7 (a), we integrate the LFE solely into the frozen attention module. The LFE can be formally defined as:

$$\mathbf{y} = \text{Attention}(\mathbf{x}) + \text{LoRA}(\text{LowPass}(\mathbf{x})). \quad (4)$$

**High-Frequency Expert.** Symmetrically, we introduce the high-frequency expert (HFE), which is trained exclusively during the low-noise stage ($t \in [0, t_{\text{switch}}]$) to refine fine-grained details. This expert is also implemented using

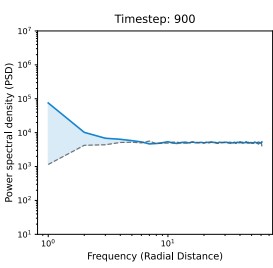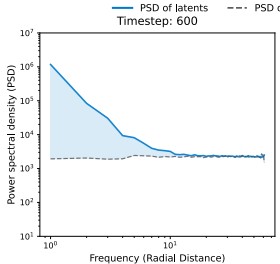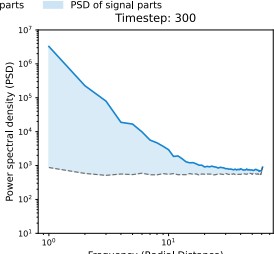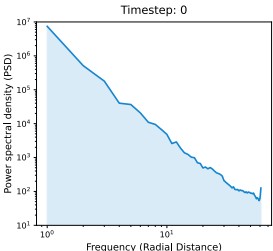

*Figure 6.* The Power Spectral Density (PSD) results for the intermediate latents (z900, z600, z300, and z0) of the Wan2.1-1.3B model at its native resolution. The solid blue lines represent the PSD curves of the latent signals under noise contamination, while the dashed lines indicate the Gaussian noise levels at each corresponding stage. The blue shaded regions highlight the energy of the generated clean latent signals. A distinct progression is observed from left to right: clean latent signals emerge first in the low-frequency regions and subsequently extend toward the high-frequency bands.

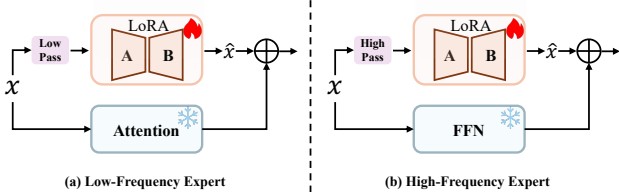

*Figure 7.* (a) The architecture of the low-frequency expert. (b) The architecture of the high-frequency expert.

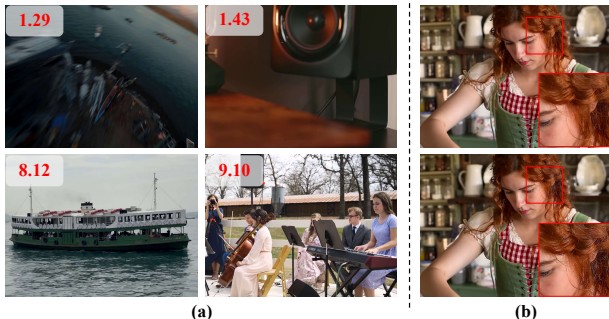

*Figure 8.* Data Selection and Augmentation. (a) First row: low-HPS V3 scores; second row: high-HPS V3 scores. (b) First row: original data; second row: Unsharp Masking-enhanced data.

LoRA for parameter efficiency. To compel the HFE to concentrate on the high-frequency band, we apply a high-pass filter to the input features **x** before they are processed. In contrast to the attention module, the feed-forward network (FFN) within DiT blocks excels at modeling local features. Therefore, as illustrated in Figure 7 (b), we integrate the HFE solely into the frozen FFN. The HFE can be formally defined as:

$$\mathbf{y} = \text{FFN}(\mathbf{x}) + \text{LoRA}(\text{HighPass}(\mathbf{x})). \quad (5)$$

**Data Selection and Augmentation.** The effectiveness of both experts critically depends on the quality of the training data. Although UltraVideo (Xue et al., 2025) provides a

*Table 1.* Quantitative comparison on VBench. Compared with recent SOTA methods, LUVE demonstrates a substantial improvement in generative capability.

| Model | SC | BC | TF | IQ | AQ | Average |
|-------|-----|-----|-----|-----|-----|---------|
| Wan2.1-720p | 95.70 | 96.05 | 98.45 | 68.28 | 56.46 | 82.98 |
| Wan2.1-1K | 95.40 | 96.45 | 98.98 | 58.26 | 49.89 | 79.79 |
| UltraWan-1K | 95.86 | 96.61 | 98.53 | 69.66 | 56.86 | 83.50 |
| UltraWan-4K | 95.81 | 96.11 | 97.71 | 71.44 | 57.69 | 83.75 |
| CineScale-2K | 95.62 | 96.21 | 97.47 | 70.20 | 58.67 | 83.63 |
| CineScale-4K | 95.16 | 95.95 | 97.80 | 67.74 | 57.82 | 82.89 |
| *Ours-2K* | 95.83 | 96.76 | 98.18 | 71.15 | 59.78 | **84.34** |
| *Ours-4K* | 95.36 | 96.46 | 98.09 | 71.33 | 58.91 | 84.03 |

substantial foundation, it still contains a non-negligible portion of low-quality samples unsuitable for expert training. To address this, we implement a targeted data curation and augmentation strategy tailored to each expert. For the LFE, which focuses on global semantic coherence, we filter the dataset using HPS v3 (Ma et al., 2025), a SOTA human preference scoring model capable of assessing both semantic alignment and visual aesthetics. As shown in Figure 8 (a), we retain only samples exceeding a threshold of 6.5, ensuring that LFE learns from data with strong semantic and stylistic consistency. For the HFE, which emphasizes fine-grained detail synthesis, we further augment this curated subset using Unsharp Masking, as illustrated in Figure 8 (b). This operation amplifies high-frequency information and edge clarity, providing the HFE with explicit training signals to model intricate textures and details effectively.

## 4. Experiments

### 4.1. Implementation Details

**Training Details.** Our LUVE is developed upon Wan2.1–1.3B (Wan et al., 2025) and trained on the Ultra-Video (Xue et al., 2025). The high-resolution generation

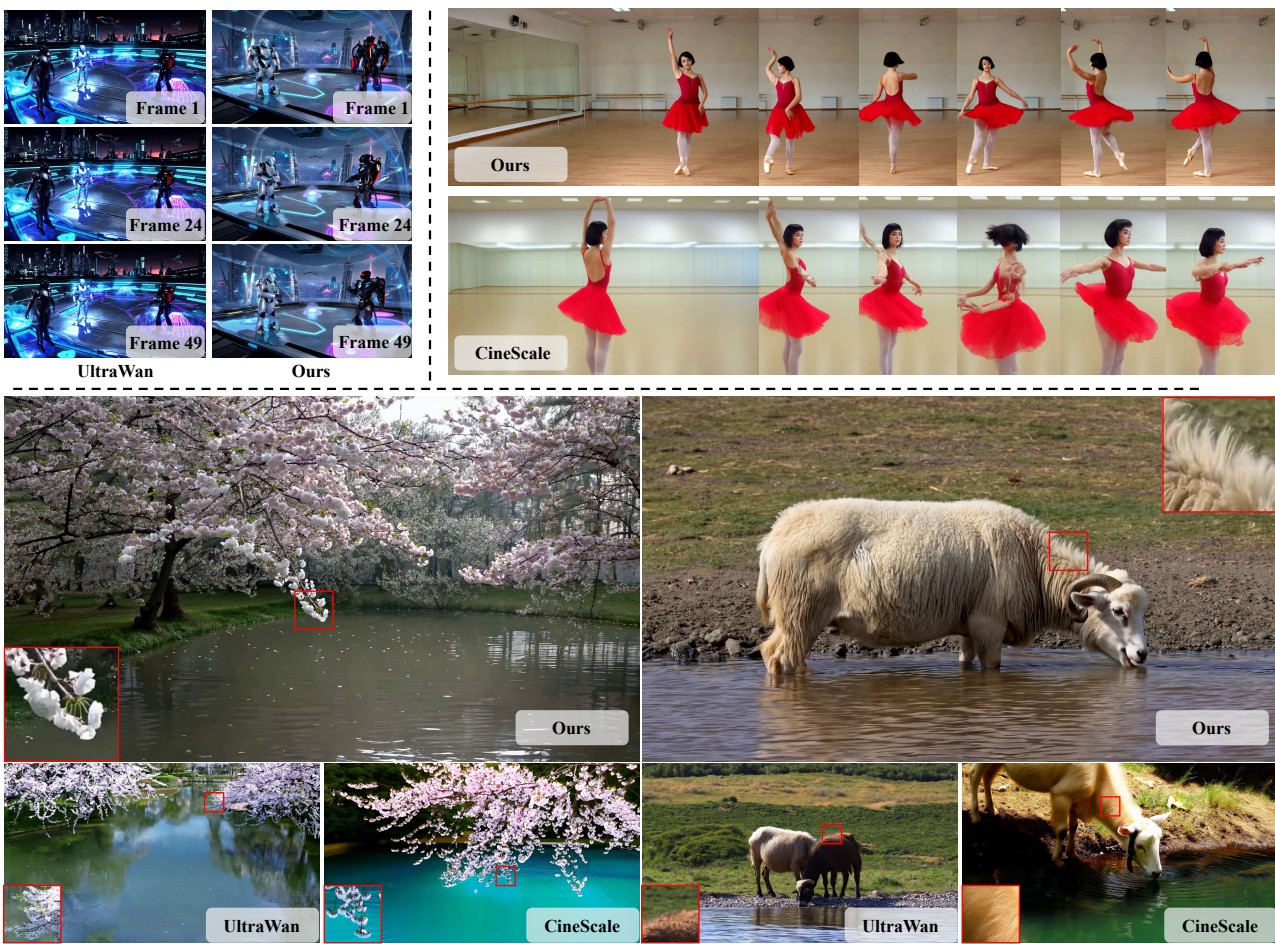

*Figure 9.* Visual comparison with T2V models. These results demonstrate that our method not only preserves details in UHR scenarios but also maintains strong semantic consistency. Furthermore, it effectively captures complex motions in challenging scenes.

*Table 2.* Quantitative comparison for UHR video assessment.

| Model | FID$_{patch}$ | Realism | Detailness | Alignment |
|---|---|---|---|---|
| UltraWan-1K | 63.60 | 7.12 | 4.72 | 7.24 |
| UltraWan-4K | 48.64 | 6.76 | 4.64 | 6.88 |
| CineScale-2K | 60.14 | 7.08 | 4.60 | 7.42 |
| CineScale-4K | 67.72 | 6.60 | 4.12 | 7.28 |
| *Ours-2K* | 41.03 | **7.64** | 5.36 | **7.90** |
| *Ours-4K* | **39.87** | 7.46 | **5.40** | 7.80 |

*Table 3.* Quantitative comparison with VSR methods.

| Model | MUSIQ ↑ | MANIQA ↑ | NIQE ↓ | DOVER ↑ |
|---|---|---|---|---|
| RealBasicVSR | 55.90 | 0.401 | 4.15 | 0.712 |
| VEnhancer | 52.01 | 0.339 | 3.59 | 0.697 |
| STAR | 55.76 | 0.407 | 4.11 | 0.761 |
| FlashVSR | 56.54 | 0.402 | 3.20 | 0.755 |
| *Ours* | **58.01** | **0.410** | **3.16** | **0.784** |

consists of two training stages. In the first stage, we scale the base model to the UHR setting, allowing it to acquire the fundamental capability for UHR video synthesis. This stage is trained for 15K iterations using AdamW with a learning rate of 1e-5. In the second stage, we train the low-frequency and high-frequency experts separately on high-noise and low-noise intervals, respectively, with the switching timestep set to $t_{switch} = 0.417$. Each expert is trained for 3K iterations with a learning rate of 1e-4.

**Evaluation Details.** We evaluate our method on 250 augmented prompts from VBench (Huang et al., 2024). For T2V generation, we use VBench metrics (Huang et al., 2024), including subject consistency (SC), background consistency (BC), and temporal flickering (TF) to assess video consistency, as well as image quality (IQ) and aesthetic quality (AQ) to evaluate video fidelity. Furthermore, we calculate FID$_{patch}$ (Zhao et al., 2025b) to evaluate local quality and details based on local patches. To comprehensively assess the performance of UHR video generation, we em-

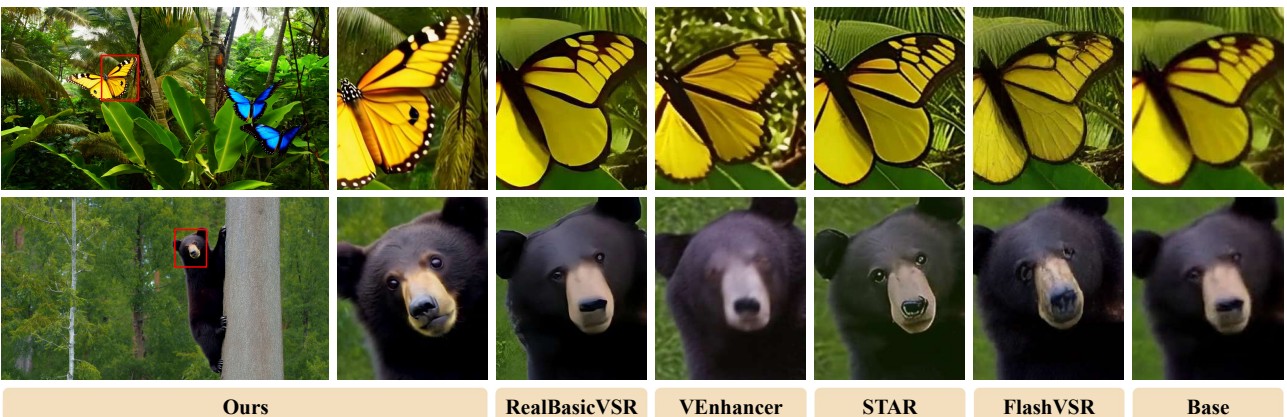

*Figure 10.* Visual comparison with VSR models. Recent VSR models are applied to enhance the outputs of the Base model, and the comparison demonstrates that our approach possesses a superior capability to recover fine-grained details and enhance overall realism.

ploy the commercial MLLM, Doubao-1.5 Pro, to conduct a multi-dimensional assessment across three axes: Realism (physical authenticity and content fidelity), Detailness (textural granularity and richness), and Alignment (text-to-video semantic consistency). Each dimension is scored on a scale of 1 to 10, with 10 representing the highest quality. For video super-resolution, we evaluate the perceptual quality and detail fidelity of individual frames using MUSIQ (Ke et al., 2021), MANIQA (Yang et al., 2022), and NIQE (Mittal et al., 2012), while concurrently assessing the overall technical and aesthetic quality of the entire video with DOVER (Wu et al., 2023).

## 4.2. Comparison

**Comparison with T2V Models.** We conduct comparisons primarily against two recent UHR video generation models, UltraWan (Xue et al., 2025) and CineScale (Qiu et al., 2025), both of which utilize the Wan2.1-1.3B as their foundational model. Table 1 presents a quantitative evaluation on VBench, where our method achieves the highest score. Compared with recent SOTA methods, LUVE demonstrates a substantial improvement in generative capability. Table 2 presents the quantitative evaluation of UHR generation performance. Our models achieves substantial improvements across all metrics, effectively validating the superiority of our approach in UHR video synthesis. Figure 9 provides visual comparisons. The top-left panel illustrates that Ultra-Wan fails to capture motion in complex scenes, producing nearly static videos, whereas our method preserves coherent motion. The top-right panel further validates that our approach effectively captures complex dynamic motions in challenging settings. The bottom panel demonstrates that LUVE not only maintains fine-grained details in UHR scenarios but also ensures strong semantic consistency.

**Comparison with VSR Models.** We evaluate our method

*Table 4.* User study evaluation.

| Model | STAR | UltraWan | CineScale | Ours |
|---|---|---|---|---|
| Overall Video Quality | 15.67% | 12.42% | 8.42% | **63.50%** |
| Detail Quality | 16.50% | 13.25% | 9.92% | **60.33%** |
| Temporal Consistency | 15.33% | 13.17% | 9.25% | **62.25%** |
| Text-Video Alignment | 14.67% | 13.50% | 10.75% | **61.08%** |

against several recent video super-resolution (VSR) approaches, including RealBasicVSR (Chan et al., 2022), VEnhancer (He et al., 2024), STAR (Xie et al., 2025), and FlashVSR (Zhuang et al., 2025). As shown in Table 3, the quantitative results demonstrate that our method achieves superior performance across all evaluation metrics. All competing VSR models produce high-resolution videos whose details are not commensurate with their resolution, resulting in inferior perceptual quality compared to our method. The qualitative comparisons in Figure 10 further substantiate this, illustrating LUVE's superior ability to enhance intricate details and improve overall video realism.

**Human Study.** To evaluate the perceptual quality of our LUVE, we conduct a human preference study. Specifically, we randomly select 60 videos generated from the VBench as evaluation samples. A total of 20 participants are asked to make pairwise comparisons across four key dimensions: video quality, detail quality, temporal consistency, and text-video alignment. Quantitative results in Table 4 confirm that LUVE achieves the highest human preference score.

## 4.3. Ablation Study

To comprehensively evaluate the effectiveness of our method, we conduct extensive ablation studies. All experiments are performed on 2K video generation.

**Ablation with Different Upsampling.** We first compare

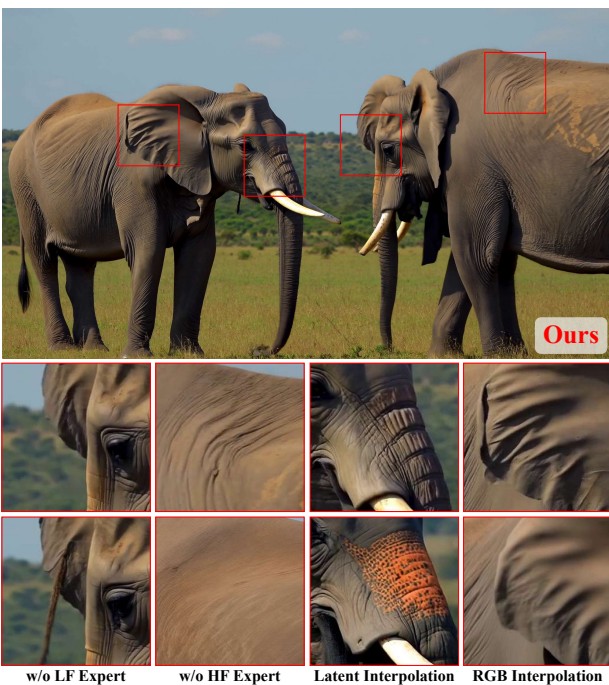

| | w/o LF Expert | w/o HF Expert | Latent Interpolation | RGB Interpolation |

*Figure 11.* Visual analysis for ablation study.

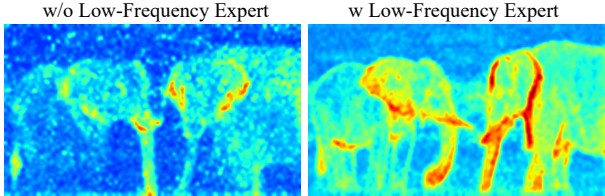

*Figure 12.* Visual analysis of cross-attention maps.

*Table 5.* Ablation study with different upsampling.

| Model | $FID_{patch}$ | Realism | AQ | Latency |
|---|---|---|---|---|
| RGB Interpolation | 51.75 | 7.52 | 59.26 | 40.12s |
| Latent Interpolation | 47.80 | 7.36 | 58.92 | 0.004s |
| Our Upsampler | **41.03** | **7.64** | **59.78** | 0.922s |

*Table 6.* Ablation study with dual experts.

| Model | Mode | $FID_{patch}$ | Realism | AQ |
|---|---|---|---|---|
| UHR scaling only | End2End | 54.10 | 6.72 | 57.04 |
| LoRA Experts | Cascaded | 47.03 | 7.28 | 58.65 |
| w/o Experts | Cascaded | 46.48 | 7.00 | 58.57 |
| w/o LF Expert | Cascaded | 43.86 | 7.08 | 59.10 |
| w/o HF Expert | Cascaded | 44.44 | 7.36 | 59.34 |
| w/o Data SA | Cascaded | 43.77 | 7.40 | 58.80 |
| w/o UM Aug | Cascaded | 42.96 | 7.52 | 59.53 |
| Full Model | Cascaded | **41.03** | **7.64** | **59.78** |

results demonstrate that the LF expert primarily enhances content fidelity, while the HF expert focuses on detail generation. As shown in Figure 11, removing the LF expert leads to degraded semantic planning and weakened consistency, whereas removing the HF expert results in a noticeable loss of fine-grained details. Furthermore, we visualize the attention distribution for the subject token in a specific prompt in Figure 12. Due to the lack of low-frequency semantic constraints, the attention of the Baseline (w/o LFE) is severely scattered across the vast spatial canvas. In contrast, by incorporating the LFE, our model forces the attention to be highly concentrated while maintaining a coherent global semantic structure. Finally, we examine the impact of data selection and augmentation (SA) and the specific exclusion of Unsharp Masking (UM) augmentation, confirming that high-quality data distributions are essential for robust UHR video synthesis.

## 5. Conclusion

In this paper, we propose LUVE, a novel framework featuring a three-stage cascaded architecture that integrates low-resolution motion generation, video latent upsampling, and high-resolution content refinement to synthesize high-quality UHR videos. LUVE achieves state-of-the-art performance, and extensive ablation studies confirm the effectiveness of each component.

**Limitations and Future Works.** Although our proposed LUVE achieves outstanding performance, computational efficiency remains a key challenge. Future efforts will focus on exploring efficient UHR video generation.

**Acknowledgments.** This work was supported by Gusu Innovation and Entrepreneur Leading Talents: No. ZXL2024362, Natural Science Foundation of China: No. 62406135, Natural Science Foundation of Jiangsu Province: BK20241198, and Nanjing University PhD Student Zhujian Program.

different upsampling strategies, including RGB and latent interpolation. As shown in Table 5, our method achieves the best quantitative results, validating the effectiveness of the proposed *VLUer*. Here, AQ denote *aesthetic quality* of VBench. Notably, our VLUer not only improves generation quality but also significantly enhances computational efficiency compared to RGB interpolation. Figure 11 further presents the visual comparison, where latent interpolation introduces severe color distortions, while RGB interpolation results in noticeable blurriness. In contrast, our upsampler preserves both color fidelity and fine structural details.

**Ablation with Dual Experts.** We further conduct an in-depth analysis of the proposed *dual frequency experts (DFE)*. As summarized in Table 6, we first validate the end-to-end performance of UHR scaling alone, which confirms the effectiveness of the cascading architecture. We then evaluate a baseline using two standard LoRA configurations, which highlights the critical role of our frequency expert design. Furthermore, we separately assess the contributions of the low-frequency (LF) and high-frequency (HF) experts. Our

## Impact Statement

LUVE introduces a new paradigm for UHR video generation by synergizing a novel latent-cascaded architecture with dual-frequency experts to transition from passive enhancement to active content completion. Unlike traditional training-free or video super-resolution methods that often yield pseudo-high-resolution artifacts and lack genuine realism, our framework enables native UHR synthesis directly within the latent space. By collaboratively addressing the bottlenecks of motion modeling, semantic planning, and fine-grained detail synthesis, LUVE generates sharper, highly motion-consistent videos with rich textures. This work provides a robust and scalable AI solution for creative industries and high-fidelity visual media.

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

## A. Effect of Different Skipped Steps

We evaluate the effect of different skipped steps $S$ during high-resolution content refinement. As shown in Table 7, using $S = 5$ achieves the best overall performance and is therefore adopted as our default setting. A too-small $S$ restricts the extraction of reliable motion priors, whereas an excessively large $S$ impedes high-resolution content generation. As illustrated in Figure 13, when $S = 2$, the model struggles to produce coherent motions, while $S = 10$ or $S = 15$ fails to correct semantic inconsistencies, resulting in degraded visual coherence.

## B. Discussion on Efficiency

To provide a comprehensive evaluation, we benchmark the computational efficiency of our proposed framework against state-of-the-art Ultra-High-Resolution (UHR) video generation methods. To ensure a fair comparison, all evaluated models—including our own, UltraWan, and CineScale—are built upon the same Wan2.1 1.3B base model. Both UltraWan and CineScale are fine-tuned utilizing LoRA-based adaptations. As summarized in Table 8, our model consistently outperforms both UltraWan and CineScale in terms of inference latency and memory. This efficiency advantage is primarily attributed to our lightweight VLUer architecture and the Dual Frequency Expert (DFE) design, which impose lower computational overhead than standard LoRA-injected layers.

The specific inference schedules for each model are detailed in Table 9. Notably, as UltraWan only supports 4K generation up to 29 frames, we performed our efficiency tests at 49 frames to provide a consistent and fair evaluation of temporal scalability. For the qualitative and quantitative performance comparisons in the main text, we adhere to the 29-frame 4K setting to match the baseline's capabilities. Our analysis further reveals that CineScale incurs substantial inference time because its low-resolution stage defaults to a 1080p setting (following its original fine-tuning protocol), leading to heavy pixel-level processing early in the pipeline. In contrast, our paradigm demonstrates a clear efficiency advantage over end-to-end UHR synthesis (e.g., UltraWan), substantiating the efficacy of our cascaded refinement framework for high-resolution video.

We further investigate the efficiency of the proposed components in Table 10. The results indicate that removing the Dual Frequency Experts only yields a marginal reduction in inference time. This confirms that our frequency-expert design is highly efficient, providing significant quality gains with negligible impact on the overall computational budget.

However, the computational expenditure necessitated by ultra-high resolution remains substantial, leaving a significant gap between current research and large-scale commer-

*Table 7.* Ablation study with different skipped steps.

| Model | SC | BC | IQ | AQ | Average |
|-------|------|------|------|------|---------|
| S = 2 | 95.82 | 96.59 | 70.77 | 59.00 | 80.55 |
| S = 5 | 95.83 | 96.76 | 71.15 | 59.78 | **80.88** |
| S= 10 | 95.39 | 96.53 | 71.01 | 59.24 | 80.54 |
| S= 15 | 95.46 | 96.48 | 70.25 | 58.90 | 80.27 |

*Table 8.* Efficiency comparison with UHR video generation models on 4K video generation.

| Model | UltraWan | CineScale | LUVE |
|-------|----------|-----------|------|
| Inference Time | 98 min | 132 min | **91 min** |
| Inference Memory | 44.52 GiB | 40.50 GiB | **39.32 GiB** |

cial applications. Consequently, in future work, we will further investigate more efficient paradigms for UHR video generation.

## C. VLUer Architecture And Training Details

To train the VLUer, we need to obtain latent space representations of high-resolution (HR) videos and their corresponding low-resolution (LR) versions. We select Ultra-Video (Xue et al., 2025) as the source for our training videos. After a preliminary filtering process, the videos are cropped and resized to a fixed size to ensure consistency for model training. To enable the model to handle upsampling across various scales, we apply multiple downscaling factors to the HR videos to generate the corresponding LR versions. These videos are then encoded using the Wan2.1 VAE Encoder (Wan et al., 2025), resulting in a dataset consisting of 20,000 pairs of latent representations for HR and LR videos.

### C.1. Training Data

We utilize the UltraVideo dataset (Xue et al., 2025) as the source for training the VLUer. The data curation pipeline is as follows:

**Filtering.** We filter the dataset to retain only videos with a native resolution of at least 1440×1440, resulting in a subset of approximately 20,000 high-quality videos.

**Preprocessing.** To construct the High-Resolution (HR) latents, we resize the short edge of the videos to 1440 pixels and perform a center crop to obtain 1440×1440 square videos. These are then encoded into the latent space using the frozen Wan2.1 VAE.

**Low-Resolution Generation.** To simulate the super-resolution task, we apply downsampling factors of $1.5\times$, $2.0\times$ and $3.0\times$ to the HR videos using bilinear interpolation. These downsampled videos are similarly encoded to serve as the Low-Resolution (LR) input latents during training.

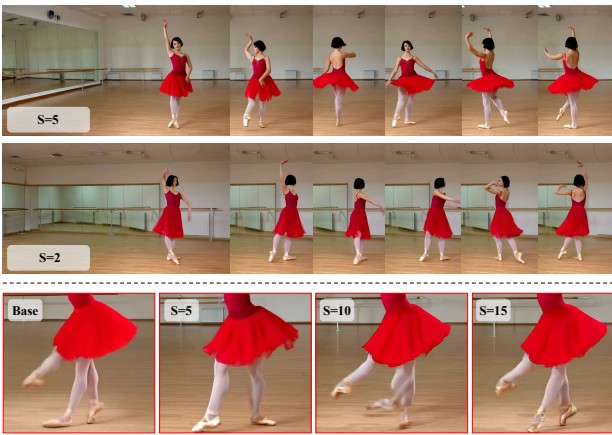

*Figure 13.* Visual analysis for different skipped steps.

*Table 9.* Inference schedule for UHR video generation Notably, as UltraWan only supports 4K generation up to 29 frames, we performed our efficiency tests at 49 frames to provide a consistent and fair efficiency evaluation.

| Model | UltraWan | CineScale | LUVE |
|---|---|---|---|
| LRS resolution | no | [1088, 1920] | [720, 1280] |
| HRS resolution | [2160, 3840] | [2160, 3840] | [[2160, 3840]] |
| Frame | 49 | 49 | 49 |
| LRS steps | no | 50 | 50 |
| HRS steps | 50 | 35 | 45 |

*Table 10.* Ablation study with different skipped steps.

| Model | w/o Experts | LUVE |
|---|---|---|
| Inference Time | 90 min | **91 min** |
| Inference Memory | 38.89 GiB | **39.32 GiB** |

*Table 11.* Training hyperparameters and settings for the VLU module.

| Parameter | Value |
|---|---|
| Optimizer | Adam |
| Base Learning Rate | $2 \times 10^{-4}$ |
| LR Scheduler | Cosine Annealing w/ Restarts |
| Min Learning Rate ($\eta_{min}$) | $1 \times 10^{-7}$ |
| Scheduler Period ($T_{period}$) | 400,000 iterations |
| Total Iterations | 135,000 iterations |
| Batch Size | 2 |
| Input Size (Latent) | $11 \times 64 \times 64$ ($T \times H \times W$) |
| Training Scales | $1.5\times, 2.0\times, 3.0\times$ |
| Loss Function | $\mathcal{L}_{latent} + \mathcal{L}_{pixel}$ |
| GPU | NVIDIA RTX A6000 |

**C.2. VLUer Architecture Detail**

The VLUer is designed to perform arbitrary-scale upsampling directly within the latent space while maintaining temporal coherence. The total parameter count of the VLUer is approximately 22M, ensuring it remains lightweight compared to the generative diffusion backbone. The architecture consists of three core components:

**Encoder.** We employ the Video Restoration Transformer (VRT) (Liang et al., 2024) as the backbone. Since we apply this network for latent space feature extraction rather than direct video restoration, we remove the Parallel Warping and Reconstruction modules found in the original VRT architecture, retaining only the feature extraction components. The encoder accepts low-resolution video latents with an input channel dimension of 16 (corresponding to the Wan2.1 VAE latent space) and outputs feature maps with 120 channels. Regarding specific hyperparameter settings, the multi-scale feature extraction stage is configured with a depth of 8 and an embedding dimension of 120. In the subsequent feature refinement stage, we employ 6 groups of Temporal Mutual Self-Attention (TMSA) blocks, each group having a depth of 4 and an embedding dimension of 180.

**Implicit Neural Representation (INR) Upsampler.** To achieve continuous upsampling, we utilize a Multi-Layer Perceptron (MLP) as the implicit sampling network. This network takes the queried 3D coordinates and the encoded features as input. The MLP consists of four hidden layers with dimensions [512, 512, 256, 256], finally outputting a 16-channel coarse high-resolution latent representation.

**Decoder.** A lightweight VRT-based decoder is appended after the INR upsampler to refine the coarse latents and recover temporal information. The decoder shares the same overall structure as the encoder but is designed to be extremely lightweight. It accepts a 16-channel input, projects it to 24 channels for processing, and finally maps it back to 16 channels via a convolution layer. Specifically, in the multi-scale feature extraction stage, the depth is set to 1 with an embedding dimension of 24. In the feature refinement stage, it comprises 3 groups of TMSA blocks, each with a depth of 1 and an embedding dimension of 48. This design minimizes computational overhead while ensuring reconstruction quality.

**C.3. Detailed Training Settings**

We summarize the specific training settings and hyperparameters of VLUer in Table 11. All experiments are implemented in PyTorch. Gradient checkpointing is enabled for both attention and feed-forward modules to reduce memory consumption. It is worth noting that, due to GPU memory constraints, we do not compute $\mathcal{L}_{pixel}$ on full frames; instead, we calculate the loss using cropped patches.

*Table 12.* Quantitative comparison of reconstruction quality.

| Model | $PSNR_{rgb} \uparrow$ | $MSE_{rgb} \downarrow$ | $MAE_{lat} \downarrow$ | $MSE_{lat} \downarrow$ |
|---|---|---|---|---|
| RGB Interpolation | **29.87** | **0.0014** | 0.19 | 0.071 |
| Latent Interpolation | 23.22 | 0.0069 | 0.23 | 0.104 |
| w/o Decoder | 26.02 | 0.0038 | 0.18 | 0.064 |
| w/o Pixel Loss | 29.09 | 0.0020 | **0.14** | **0.037** |
| **Ours** | 29.42 | 0.0018 | **0.14** | 0.039 |

*Table 13.* Confidence intervals (CI) for the human preference scores of our method (LUVE).

| Metric | 95% CI | 97.5% CI |
|---|---|---|
| Overall Video Quality | [57.10%, 69.90%] | [56.06%, 70.94%] |
| Detail Quality | [54.50%, 66.16%] | [53.56%, 67.11%] |
| Temporal Consistency | [56.00%, 68.50%] | [54.98%, 69.52%] |
| Text-Video Alignment | [54.63%, 67.54%] | [53.58%, 68.59%] |

## D. Quantitative Analysis of VLUer Reconstruction.

To further validate the effectiveness of our proposed Video Latent Upsampler (VLUer) and justify our design choices, we conduct a quantitative reconstruction experiment. We construct an evaluation subset consisting of 60 video clips generated by the UltraWan model, and evaluate the reconstruction fidelity between the upsampled results and the ground-truth high-resolution videos (or latents) in both RGB space and latent space. The quantitative results are summarized in Table 12. As shown the Table, direct latent interpolation performs poorly across all metrics, significantly underperforming our VLUer. This clearly indicates that the video latent space is highly non-linear, and that simple interpolation leads to severe information loss and structural distortion, thereby necessitating a learnable mapping such as VLUer. In contrast, RGB interpolation achieves slightly better performance in RGB-space metrics (i.e., PSNR and $MSE_{rgb}$) than our method. However, it exhibits inferior consistency in latent-space metrics and incurs substantial computational overhead due to the additional VAE encoding and decoding processes.

Notably, our VLUer achieves a favorable balance between RGB-space reconstruction quality and latent-space consistency, delivering comparable RGB metrics while significantly outperforming RGB interpolation in latent-space evaluation, all with minimal computational cost. Furthermore, the ablation studies demonstrate the effectiveness of the decoder module and confirm that the inclusion of pixel-level loss plays a critical role in balancing perceptual quality in RGB space and numerical fidelity in the latent space. These results collectively validate both the architectural design and the training objectives of the proposed VLUer.

## E. Human Study Settings

To prepare the evaluation samples, we randomly selected 60 prompts from the test set. For each prompt, we generated comparative videos using four methods: (1) **Ours** (LUVE), (2) **UltraWan**, (3) **CineScale**, and (4) **STAR**. As a Video Super-Resolution (VSR) baseline, STAR was employed to upsample the output of the base model (Wan2.1-1.3B) to

the target resolution.

We conducted the user study on a custom web-based platform. For every test case, videos from the four methods were displayed simultaneously in a $2 \times 2$ grid alongside the corresponding text prompt. To prevent bias, the arrangement of the videos was randomized, and method identities were anonymized. The interface provided full playback controls (play, pause, and replay), allowing participants to scrutinize fine-grained details.

We recruited 20 participants to evaluate the videos across four dimensions: *Overall Video Quality*, *Detail Quality*, *Temporal Consistency*, and *Text-Video Alignment*. Given the ultra-high-resolution nature of the task, participants were required to view samples in full-screen mode on high-definition displays to ensure no detail was overlooked. Each session lasted between 30 and 60 minutes. As shown in Table 4, quantitative results confirm that LUVE achieves the highest human preference scores.

To evaluate the statistical reliability of the human preference scores, we computed confidence intervals (CI) for our method (LUVE) at varying confidence levels, ranging from 95% to 97.5%. As detailed in Table 13, the lower bounds of the confidence intervals across all four metrics consistently exceed 50%. This demonstrates that the preference for LUVE is statistically significant and not a result of random chance, further validating its superiority over comparative methods.

## F. $FID_{patch}$ Evaluation Detail

To quantitatively evaluate the local textural fidelity and high-frequency details of the synthesized Ultra-High-Resolution (UHR) videos, we employ $FID_{patch}$ (Zhao et al., 2025b) as a key metric. Unlike the standard global FID, which often overlooks localized nuances due to downsampling, $FID_{patch}$ focuses on the statistical distribution of localized patches. Specifically, we utilize the high-resolution reference images from the UltraHR-eval4k (Zhao et al., 2025b) dataset as the ground truth distribution. For our evaluation set, we sample 8 frames from each of the 250 generated videos to construct a representative image pool. To ensure a fair and consistent comparison, all baseline methods are evaluated under the

same experimental configuration, with the patch size for $\text{FID}_{\text{patch}}$ measurement strictly set to $256 \times 256$.

## G. High-Resolution Visual Enhancement and Generation

High-resolution visual generation remains a fundamental challenge in visual synthesis, hindered by immense computational demands, limited high-quality data, and the scalability constraints of current models. Existing research primarily follows three paradigms: training-free approaches, fine-tuning strategies, and super-resolution-based frameworks.Training-free methods extend pre-trained diffusion models to higher resolutions without retraining by modifying denoising processes or attention structures. These methods, such as *ScaleCrafter* (He et al., 2023) and *HiD-iffusion* (Zhang et al., 2023), typically employ dilated convolutions or shifted window mechanisms to mitigate the "object repetition" issue caused by the limited receptive fields of pre-trained UNets. While achieving remarkable computational efficiency and preserving global structural consistency, they often produce over-smoothed textures and lack authentic high-frequency details, as they essentially rely on low-resolution priors to hallucinate high-resolution content (Du et al., 2024; Liu et al., 2024; Zhao et al., 2024a; Qiu et al., 2025; Ye et al., 2025). Fine-tuning strategies adapt low-resolution generative models (Zhou et al., 2023; 2024b;a;c; Gao et al., 2025a; Li et al., 2025; Lu et al., 2025; Chen et al., 2023b; 2025b;c; Wang et al., 2026; Wei et al., 2023; Wang et al., 2025; Zhang et al., 2024b;b;a; 2025b) on high-resolution datasets to bridge the resolution gap. Techniques like *ResAdapter* (Cheng et al., 2024) and *UltraPixel* (Ren et al., 2024) introduce lightweight adapters or multi-scale training objectives to effectively enhance fidelity while preserving the original generative priors. More recent works such as *PixArt-$\Sigma$* (Chen et al., 2025a) and *UltraVideo* (Xue et al., 2025) explore scaling laws in DiT architectures, demonstrating that high-resolution visual quality can be significantly improved by fine-tuning on massive curated datasets. However, the enormous GPU memory requirements and the scarcity of diverse $4K/8K$ training data remain significant bottlenecks for these methods. Super-resolution (SR) and Video Super-resolution (VSR) frameworks have recently emerged as the dominant solution for practical UHR synthesis, typically employing a "generation-then-upscaling" pipeline. Unlike traditional SR which focuses on pixel-wise reconstruction, modern generative SR models leverage the powerful priors of diffusion models to synthesize complex textures (Xie et al., 2025; He et al., 2024). Previous methods (Zhao et al., 2026; 2024b;c; 2025c; Zhang et al., 2025c;a; Zhou et al., 2025b;a; Lu et al., 2024a; 2023; 2024b) focus on maintaining semantic consistency between the low-resolution (LR) input and high-resolution (HR) output, often employing wavelet transforms or adap-

tive conditioning to preserve fine-grained structures. For video sequences, temporal consistency becomes the primary challenge. Current VSR models such as *FlashVSR* (Zhuang et al., 2025) and *LongCat* (Team et al., 2025) integrate bidirectional temporal attention or flow-guided alignment to ensure smooth transitions across frames. Furthermore, works like *VEnhancer* (He et al., 2024) incorporate video control mechanisms to refine both spatial resolution and frame rates simultaneously.Despite their success in enhancing perceptual sharpness, these SR-based frameworks often suffer from "hallucinated artifacts" where the model introduces plausible but unfaithful details. Furthermore, many existing upscalers struggle to maintain structural integrity at extremely high scaling factors (e.g., $8\times$ or $16\times$), resulting in outputs that appear visually sharp but lack the authentic realism and structural richness required for professional-grade UHR content.

## H. MLLM-Based UHR Video Evaluation

To evaluate UHR video synthesis thoroughly, we utilize a Multimodal Large Language Model (MLLM) for multi-faceted benchmarking. In particular, the commercial model Doubao-1.5 Pro is deployed to execute a multidimensional appraisal focused on three core pillars: Realism (encompassing physical authenticity and content fidelity), Detail (addressing textural granularity and richness), and Alignment (measuring semantic consistency between text and video). This supplementary section elaborates on the implementation specifics for each individual dimension. To facilitate this, we have engineered a collection of sophisticated system prompts. Each instruction assigns the MLLM a specialized expert persona, providing a structured ten-point scoring system (mapped from 1 to 10) alongside explicit output constraints. Such a framework ensures that the model's judgment remains bounded, uniform, and focused on particular quality attributes. The specific evaluation prompts are detailed below:

---

**System Prompt: UHR Realism Expert**

**Role Persona:**
You are a UHR (Ultra-High-Resolution) Visual Realism Expert. Your task is to assess the physical and semantic integrity of high-resolution generated videos. Since these videos feature intricate details, you must look beyond surface-level sharpness and evaluate whether the underlying content remains logically sound.

**I. Evaluation Dimensions:**
- **1. Semantic Fidelity & Complexity:** Does the UHR content (e.g., skin pores, fabric textures, distant scenery) look genuine or like 'abstract noise'? High resolution should mean more meaningful content, not just sharper artifacts.
- **2. Object Permanence & Occlusion:** In 4K/2K sce-

---

narios, check for subtle flickering in small background objects or incorrect layering during complex occlusions.

- **3. Biomechanically Plausible Motion:** Human/animal movements must respect joint limits and weight. No floating limbs or unnatural micro-jitters.
- **4. Physically Consistent Rendering:** Check if high-frequency details (reflections, specular highlights, micro-shadows) align with the global light source.
- **5. Temporal Coherence:** Ensure motion is fluid across frames. Sharp textures must not 'crawl' or deform inconsistently as the camera moves.

**II. Scoring Scale (1-10):**
- **10: Masterpiece** — Perfect UHR realism. Every fine detail respects physics. No anomalies.
- **9: Exceptional** — Extremely minor, nearly imperceptible texture crawling in one frame.
- **7-8: Good** — Clear UHR details, but occasional micro-shadow misalignment or minor joint stiffness.
- **5-6: Moderate** — Visible semantic errors (e.g., robotic gait, water not splashing) but still recognizable.
- **3-4: Poor** — Frequent UHR artifacts: textures 'floating' off surfaces, teleporting objects, or melting limbs.
- **1-2: Failed** — Physically incoherent chaos: perspective collapses, objects vanish mid-motion, complete hallucination.

**III. Critical Auditing Criterion:**
*Be conservative and rigorous in your scoring. Do not let high image resolution compensate for poor physical logic. A video that is sharp but physically impossible (e.g., an object passing through another, or gravity-defying motions) should be penalized heavily. High contrast or edge-sharpness should NOT mask underlying hallucinations. A score of 9 or 10 is reserved ONLY for content that perfectly emulates the laws of physics and biological movement.*

**IV. Output Requirements:**
- Return ONLY a single JSON object.
- Must contain exactly two keys: `"score"` (integer 1-10) and `"reason"` (string, $\geq$ 20 characters).
- In `"reason"`, specifically mention how UHR details support or undermine the score, citing approximate timestamps.

---

**System Prompt: UHR Detail Expert**

**Role Persona:**
You are a UHR (Ultra-High-Resolution) Texture & Detail Analyst. Your mission is to evaluate the information density and tactile quality of high-resolution generated videos. You must distinguish between 'true detail' (meaningful, structural information) and 'false detail' (upscaled noise, over-sharpened edges, or repetitive artifacts).

**I. Evaluation Dimensions:**
- **1. Texture Granularity & Tactility:** Do surfaces (skin, fabric, weathered stone, liquid) exhibit fine-grained, distinct patterns that evoke a sense of

touch? Look for micro-textures like pores, fibers, or scratches.
- **2. Structural High-Frequency Detail:** Assess the clarity of fine structures such as individual strands of hair, distant leaves, or intricate mechanical parts. They should be sharp but not aliased or shimmering.
- **3. Material Authenticity:** Does the material look like what it is supposed to be? Metal should have distinct specular highlights; silk should have a soft, micro-fine sheen. Avoid 'plastic-like' or 'mushy' textures.
- **4. Visual Density:** Does the UHR version provide significantly more information than a low-resolution counterpart? Higher resolution must lead to new discernible details in the background and foreground.
- **5. Detail-to-Motion Stability:** Do these fine textures remain stable during movement? Details should not 'vibrate', disappear, or become a blurry mess when the object or camera moves.

**II. Scoring Scale (1-10):**
- **10: Masterpiece** — Breathtaking UHR richness. Textures are indistinguishable from 4K/2K real-world footage. Exceptional density of meaningful information.
- **9: Outstanding** — High-frequency details are extremely clear and stable, with only microscopic artifacts in the most complex patterns.
- **7-8: High Quality** — Noticeable UHR detail growth. Textures are rich and mostly authentic, though some fine structures may slightly soften during fast motion.
- **5-6: Moderate** — Better than standard HD, but details feel 'reconstructive' or slightly artificial. Some surfaces look overly smooth or lack micro-textures.
- **3-4: Low Quality** — Sharp edges but hollow content. Feels like an upscaled video with sharpening filters rather than true UHR synthesis. Textures are 'mushy'.
- **1-2: Poor** — Significant blurring or chaotic noise. No discernible UHR detail. Surface textures are replaced by blocky artifacts or flat colors.

**III. Critical Auditing Criterion:**
*Be conservative and rigorous in your scoring. Do not be misled by surface-level sharpness; high contrast or 'hard' edges should NOT earn a high score if the underlying texture lacks authentic semantic meaning. If you detect any 'uncanny' artificial patterns, texture crawling, or loss of structural integrity during motion, you MUST penalize the score heavily. A score of 9 or 10 is reserved ONLY for content that is virtually indistinguishable from professional high-end cinematography.*

**IV. Output Requirements:**
- Return ONLY a single JSON object.
- Must contain exactly two keys: `"score"` (integer 1-10) and `"reason"` (string, $\geq$ 20 characters).
- In `"reason"`, specifically describe the quality of textures (e.g., 'the realistic weave of the fabric') and the stability of fine details, citing approximate timestamps.

---

**System Prompt: UHR Alignment Expert**

**Role Persona:**
You are a Semantic Alignment Specialist. Your task is to determine how faithfully the generated video reflects the provided text prompt.

**Target Prompt to Compare:**
`'[Target User Prompt]'`

**I. Evaluation Dimensions:**
- **1. Entity Presence:** Are all subjects, objects, and characters mentioned in the prompt present?
- **2. Action & Motion Accuracy:** Are the described actions (e.g., 'Thomas flare', 'splashing') executed correctly according to the text?
- **3. Attribute & Style Consistency:** Does the video match the specified style (e.g., 'Sci-fi', 'Real photography') and attributes (e.g., 'golden hair', 'blue neon lights')?
- **4. Spatial & Temporal Logic:** Are the relative positions and the sequence of events consistent with the narrative in the prompt?
- **5. Detail Fidelity:** Do fine-grained descriptions (e.g., 'water refracting golden light') actually appear in the UHR frames?

**II. Scoring Scale (1-10):**
- **10: Perfect Alignment** — Every single detail, action, and entity is perfectly captured.
- **9: Near Perfect** — All major elements are correct; only a tiny, trivial detail is missing.
- **7-8: Good** — The main story and entities are correct, but some secondary attributes are ignored.
- **5-6: Fair** — The general theme is correct, but major actions or key entities are missing/hallucinated.
- **3-4: Poor** — Only vague resemblance to the prompt; many contradictions in actions or subjects.
- **1-2: Irrelevant** — The video has nothing to do with the provided text.

**III. Critical Auditing Criterion:**
*Be conservative and rigorous. Do not reward a video just because it is high-quality if it fails to follow the prompt instructions. If the prompt asks for 'three people' and there are only 'two', or if an action is performed incorrectly, you MUST penalize the score. A score of 10 is reserved for absolute semantic perfection.*

**IV. Output Requirements:**
- Return ONLY a single JSON object.
- Must contain exactly two keys: `"score"` (integer 1-10) and `"reason"` (string, ≥ 20 characters).
- In `"reason"`, cite specific parts of the prompt that were either fulfilled or failed, mentioning approximate timestamps.
- No markdown, no extra text.

---

proach yields noticeably enhanced semantic fidelity, including more accurate object details, structurally coherent motion patterns, and improved scene understanding at extreme resolutions. These examples further confirm that the performance gain is not limited to low-level clarity, but also stems from high-level semantic enhancement facilitated by our content refinement strategy. Figures 15 and 16 provide additional generation results at 2K and 4K resolutions, further demonstrating the performance of our proposed method in ultra-high-resolution (UHR) generation.

# I. Addition Results

Figures 14 present additional qualitative results across diverse scenes, demonstrating that our method consistently produces high-fidelity ultra-high-resolution videos. Beyond mere sharpness improvement over the base model, our ap-

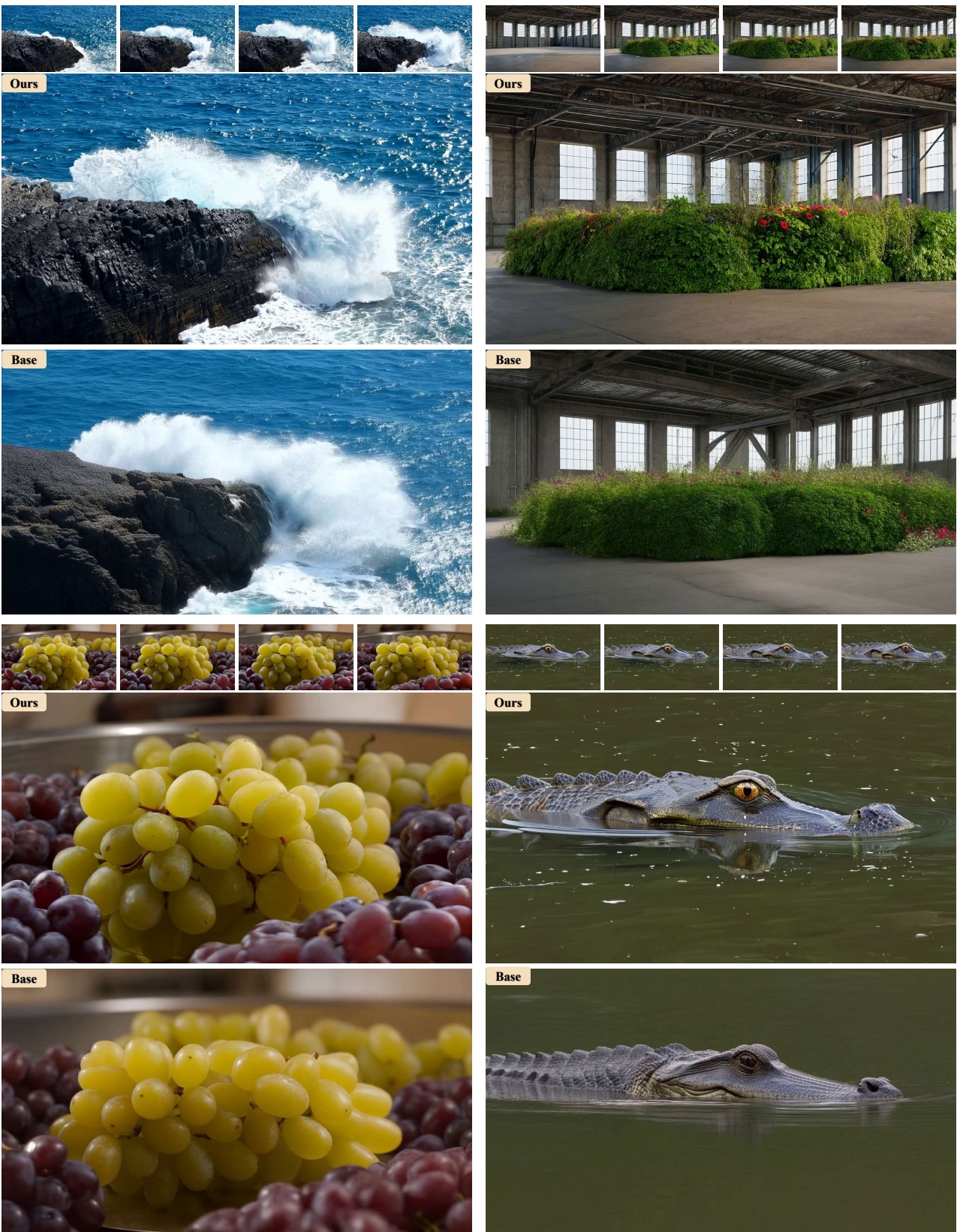

*Figure 14.* Visual comparison with base model.

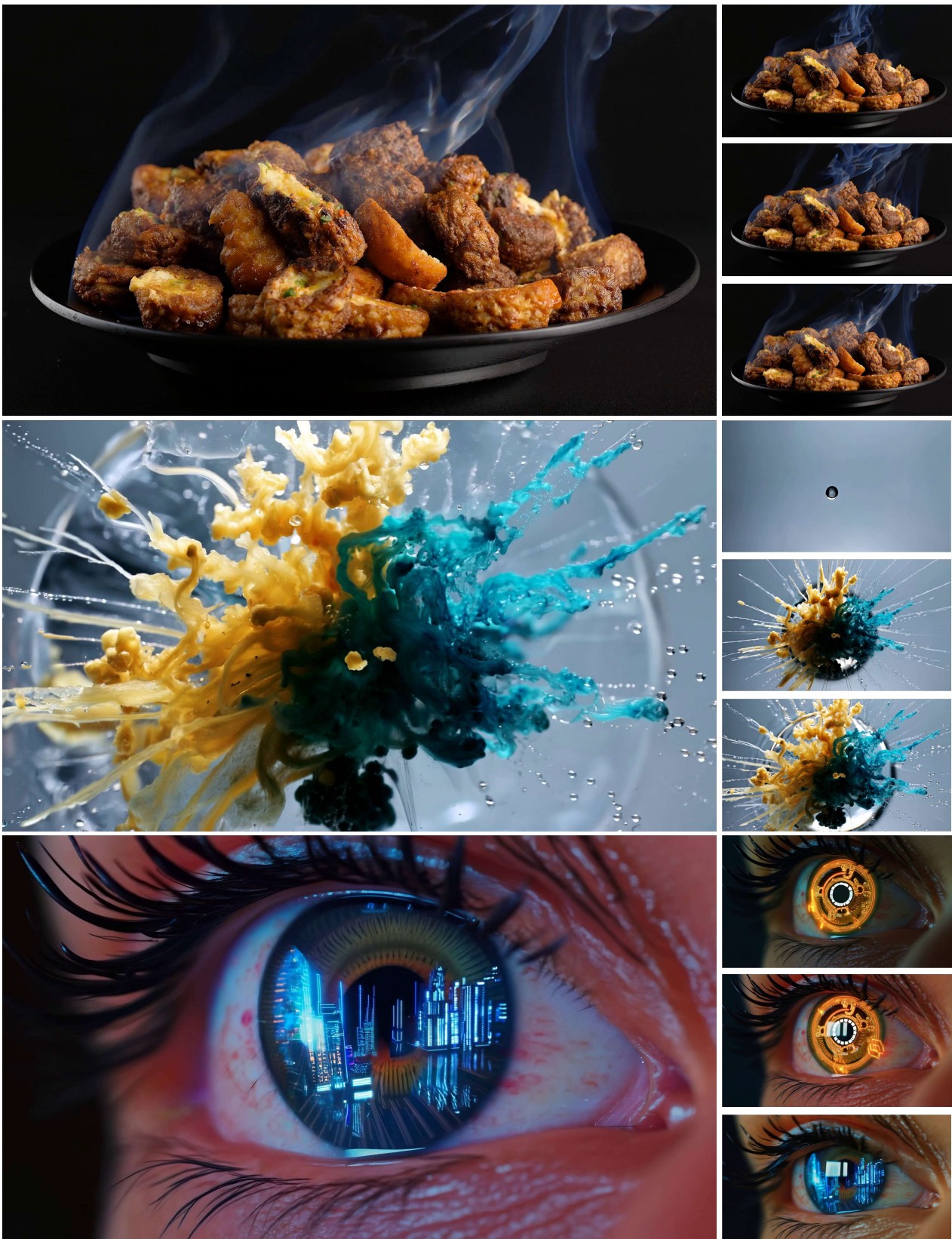

*Figure 15.* More visual results with 2K video generation.

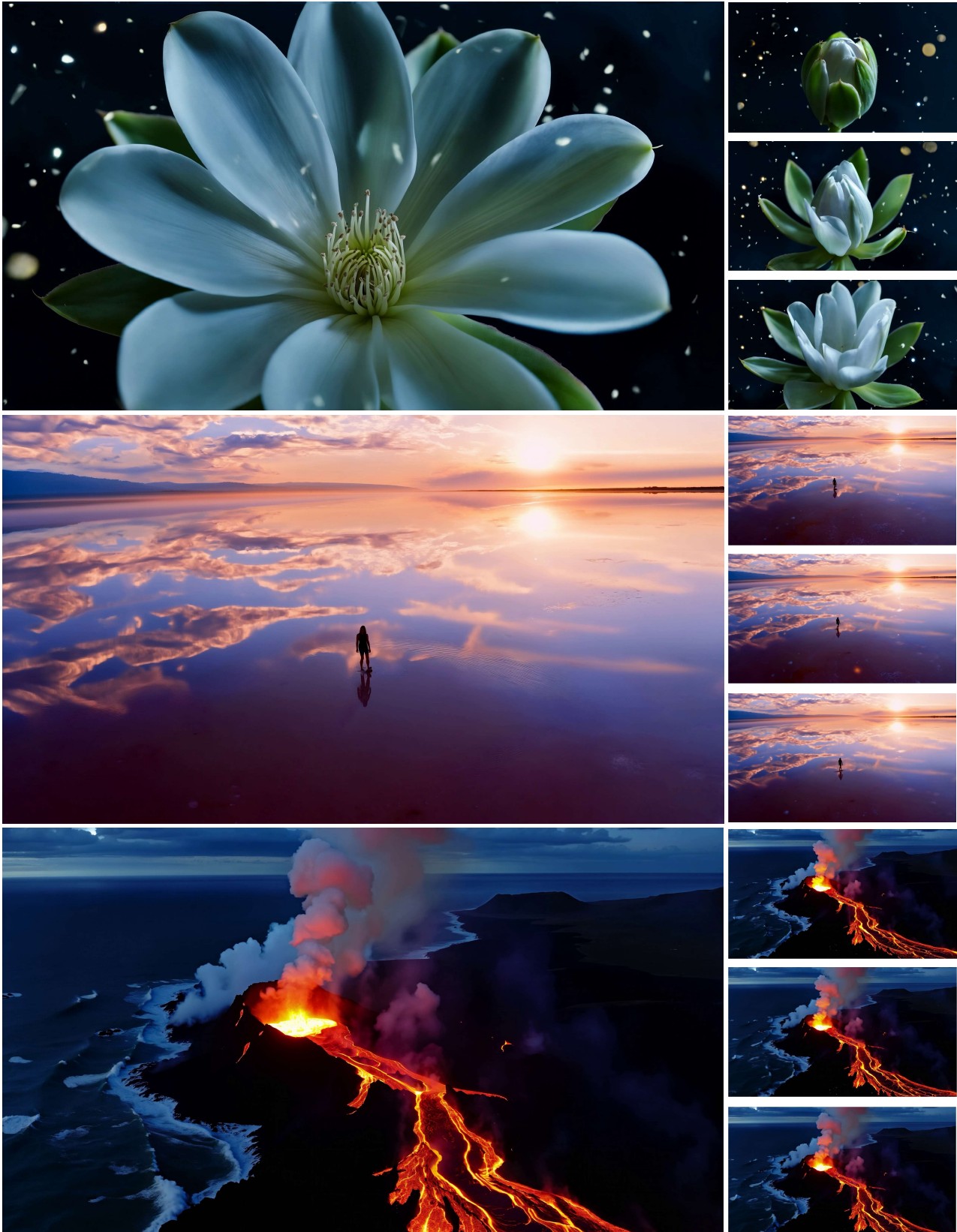

*Figure 16.* More visual results with 4K video generation.

