# OpenReview forum: "LUVE : Latent-Cascaded Ultra-High-Resolution Video Generation with Dual Frequency Experts"
_ICML.cc/2026/Conference — ICML 2026 regular_

### Official Review · Reviewer_xD9b · 2026-03-04

**Soundness:** 3
**Presentation:** 3
**Significance:** 3
**Originality:** 3
**Overall Recommendation:** 4
**Confidence:** 3

**Summary:**

This paper proposes a latent-space cascaded ultra-high-resolution (UHR) video generation framework named LUVE for generating UHR videos. Existing methods often struggle to balance motion modeling, semantic structuring, and fine-detail synthesis in this setting, especially as resolution increases. LUVE consists of three stages. First, the Low-resolution Motion Generation (LMG) module learns motion patterns at a lower resolution to establish a stable motion prior. Next, the Video Latent-space Upsampling (VLU) module upsamples within the latent space, reducing computational cost. Finally, the High-resolution Content Refinement (HCR) stage further improves visual quality: a low-frequency expert maintains semantic consistency, while a high-frequency expert synthesizes fine textures. Together, they enhance the final output. Experimental results show that LUVE outperforms existing methods on UHR video generation. Ablation studies further examine the role of each component and indicate that all three stages contribute to overall performance improvement.

**Compliance With Llm Reviewing Policy:**

Affirmed.

**Final Justification:**

I have no other questions.

**Key Questions For Authors:**

see weaknesses.

**Limitations:**

yes

**Strengths And Weaknesses:**

Strengths

1. The three-stage cascaded architecture (LMG + VLU + HCR) is well organized, with clear functional separation across modules.

2. This paper maps different frequency components to distinct diffusion noise stages, and inserts LoRA adapters into both the attention and FFN layers, which might be useful for uhd video generation.

Weaknesses
1. Despite the claim of breaking the traditional cascaded paradigm, the framework largely follows a standard pipeline that low-resolution generation, upsampling, and high-resolution refinement, combined with plug-in LoRA experts and INR-based upsampling. The contribution appears to lie more in design reconfiguration than in fundamental innovation.

2. The necessity of the frequency-specific experts is not convincingly demonstrated. It remains unclear whether the performance gains truly stem from frequency-stage coupling or from increased parameterization. The paper needs to validate and explain the claimed coarse-to-fine (low-frequency to high-frequency) modeling behavior.

3. The experimental section lacks sufficient transparency. More information about evaluation protocols, implementation specifics, and reproducibility details should be provided.

4. The computational cost such as FLOPs, number of GPUs used, and total training time are not reported.

---

> ### Author Rebuttal · Authors · 2026-03-29
>
> # W1:
> **Answer:** We sincerely thank the reviewer for recognizing the merits of our architectural design. To further clarify, we would like to re-summarize our core contributions: compared to current cascaded HR video generation paradigms, LUVE exhibits fundamental innovations across its generation paradigm, component design, and theoretical insights.
> ### **New Paradigm**:
> Current cascaded models adhere to an "LR content generation + HR detail refinement" pipeline, where the HR stage acts merely as a passive perceptual enhancer unable to rectify semantic inaccuracies or synthesize rich content. In contrast, LUVE pioneers a **new "LR motion generation + HR content refinement" paradigm**. Breaking free from passive sharpening, our HR stage functions as an active content completer, rectifying LR semantic deviations and natively synthesizing authentic UHR content.
> ### **New Component**:
> (1) Bypassing the computationally expensive VAE pixel-latent transitions, our VLUer, the first trainable video latent upsampler, operates entirely within the latent space, significantly reducing computational overhead.
> (2) Our DFE dynamically decouple generation: the LFE corrects the global semantic layout while the HFE injects UHR textures. This overcomes previous methods, achieving highly precise structural and detail synthesis.
> ### **New Insight**:
> Beyond architectural designs, LUVE provides crucial new insights: (1) Scaling to UHR should not be limited to mere perceptual enhancement. True UHR generation fundamentally advances semantic coherence, content fidelity, and visual richness rather than just improving sharpness. (2) Scaling to UHR drastically expands spatial information, overwhelming standard DiTs during their natural low-to-high frequency denoising shift. By dynamically reinforcing specific frequencies, our method mitigates both semantic inconsistencies and detail degradation.
> # W2:
> **Answer:** To investigate the frequency dynamics of video diffusion models during denoising, we perform Power Spectral Density (PSD) analysis on Wan2.1. As shown in **Fig. 1**, the model shows a clear frequency behavior: primarily capturing low-frequency components in the high-noise stage, then gradually shifting focus toward high-frequency details in the low-noise stage. However, scaling to UHR substantially expands spatial information, resulting in low-frequency semantic structural inconsistencies and high-frequency detail degradation. To examine this, we compare the PSD distributions of our full model, the baseline (w/o Experts), and the ground truth (GT) (**Figs. 2 and 3**). For a fair comparison, we use real-world UHR videos as the Ground Truth (GT). The evaluated models then generate videos using prompts derived from these GT videos via Gemini 3. Results demonstrate that in the low-noise stage, our method produces significantly stronger high-frequency responses than the baseline, aligning closer with the GT. This indicates the HFE effectively enhances UHR detail reconstruction. Moreover, although our method also achieves GT-aligned PSD distributions in the high-noise stage (governed by the LFE), PSD primarily reflects spectral magnitude rather than semantic coherence. Thus, we further analyze attention maps to understand the LFE's role in modeling global semantic structures. As shown in **Fig. 4**, ours generates object-specific and coherent attention maps, whereas discarding the LFE results in diffuse and non-specific activations. This validates that the LFE effectively resolves the challenge of capturing and modeling the global semantic layout. **For all additional figures, please refer to the anonymous link**: https://anonymous.4open.science/r/LUVE_Image-41D0/README.md. Furthermore, this improvement does not merely stem from an increase in parameterization. As demonstrated in **Table 6 of our paper**, employing **LoRA Experts under the same settings** does not yield substantial performance gains, proving that the frequency experts design is the key.
> # W3:
> **Answer:** We apologize if the implementation details were difficult to locate. We have provided comprehensive setup in our **Appendix of the paper**. Specifically, we provide inference details in Section B, training configurations of VLUer in Section C, human study settings in Section E, and evaluation protocols in Sections F and G. These sections ensure complete experimental transparency.
> # W4:
> **Answer:**  Due to the highly complex and multi-component nature of video DiT architectures, calculating FLOPs is challenging. Since our framework is fine-tuned on the Wan 2.1, the additional computational cost stems solely from the LoRA-based LF and HF experts, whose detailed parameter are provided in **Table 1**. LF and HF experts were trained for 3,000 iterations on 16 A100 GPUs with a learning rate of 1e-4, requiring approximately 63 and 78 hours.
>
> **Table 1. Detailed parameter overhead for our experts.**
> | Model | Params | LoRA rank |
> |:-|:-|:-|
> |LFE|94M|96|
> |HFE|60M|96|

---

> > ### Author Rebuttal · Reviewer_xD9b · 2026-04-04
> >
> > Thank you for the rebuttal. The authors have addressed my previous concerns, so I will maintain my original score.
> >
> > That said, I acknowledge that the novelty concerns raised by other reviewers are reasonable. I agree that the level of novelty may be borderline.

---

> > > ### Author Response · Authors · 2026-04-04
> > >
> > > We sincerely thank you for your positive evaluation and are thrilled that our previous rebuttal has fully addressed your concerns. We promise that all the supplementary experiments, and clarifications provided during this rebuttal phase will be carefully integrated into the revised manuscript to further enhance the overall quality of the paper.
> > >
> > > Regarding the concerns about novelty raised by other reviewers, we would like to take this opportunity to further clarify the core contributions of our work, hoping to resonate with your perspective. Our novelty primarily stems from two fundamental shifts in addressing UHR video generation:
> > >
> > > (1) We propose a novel "LR motion generation + HR content completion" paradigm for UHR video generation. This is driven by a core philosophy: scaling to UHR should not be limited to mere perceptual enhancement. True UHR generation fundamentally advances semantic coherence, content fidelity, and visual richness, rather than just improving sharpness. This represents a **fundamental departure** from prior paradigms and conventional viewpoints.
> > >
> > > (2) We introduce a novel Frequency Expert tailored to enhance UHR video generation. This is driven by a core insight: scaling to UHR drastically expands spatial information, overwhelming standard DiTs during their natural low-to-high frequency denoising shift. By dynamically reinforcing specific frequencies, our approach effectively mitigates both semantic inconsistencies and detail degradation. This targeted mechanism is **fundamentally different** from prior architectural designs.
> > >
> > > We completely understand that concerns regarding novelty might stem from the structural simplicity of our method. However, we believe in the elegance of a straightforward design. Through extensive experiments, we have rigorously demonstrated that this simple yet well-motivated approach drastically enhances the quality of UHR video generation. As clearly evidenced by the **demo videos provided in our supplementary materials**, our model achieves state-of-the-art results in current UHR video generation.
> > >
> > > **We are profoundly grateful for your valuable feedback, which has greatly helped us improve our work. Thank you once again for your time, constructive guidance, and support.**

---

### Official Review · Reviewer_Zhti · 2026-03-09

**Soundness:** 3
**Presentation:** 3
**Significance:** 2
**Originality:** 2
**Overall Recommendation:** 3
**Confidence:** 3

**Summary:**

The paper introduces LUVE, a Latent-Cascaded Ultra-High-Resolution (UHR) video generation framework aimed at resolving the compounded challenges of motion degradation, semantic repetition, and detail blurring commonly found when scaling diffusion models to 4K resolutions. The framework comprises three stages: (1) Low-Resolution Motion Generation (LMG) to establish motion priors; (2) Video Latent Upsampling (VLU) via a novel Implicit Neural Representation (INR)-based upsampler (VLUer) to bypass VAE decoding overhead and latent manifold distortion ; and (3) High-Resolution Content Refinement (HCR), which applies a Low-Frequency Expert (LFE) and a High-Frequency Expert (HFE) at different denoising stages to enhance semantic coherence and texture synthesis respectively .

**Compliance With Llm Reviewing Policy:**

Affirmed.

**Final Justification:**

Considering the author's rebuttal and the opinions of other reviewers, I have decided to ultimately maintain my previous score.

**Key Questions For Authors:**

refer to major weakness

**Limitations:**

yes

**Strengths And Weaknesses:**

Strengths
1、The experimental evaluation demonstrates rigorous logic, clearly delineating the Text-to-Video (T2V) and Video Super-Resolution (VSR) testing protocols . The T2V evaluation combines VBench and an MLLM for a multi-dimensional assessment ; the VSR evaluation employs no-reference metrics to compare the super-resolution efficacy against mainstream tools .
2、Achieving coherent 4K video synthesis on a 1.3B parameter base model (Wan2.1) with reduced motion degradation offers practical reference value for applied generative video synthesis.
Weaknesses
1、My primary concern lies in the fairness of the baselines regarding data curation. The paper explicitly states that the LFE training data was aggressively curated using HPS v3 (threshold > 6.5), and HFE data was augmented using Unsharp Masking . However, it is unclear if the direct T2V competitors (UltraWan, CineScale) were fine-tuned on this exact same highly curated/augmented subset. If they were trained on the raw UltraVideo dataset, the performance delta observed in Table 1 and Table 2 may be heavily conflated with data quality rather than architectural superiority. The ablation in Table 6 shows a significant FID penalty when these data strategies are removed, heightening this concern.
2、Equations 4 and 5 utilize arbitrary LowPass(x) and HighPass(x) operations. The paper completely omits the mathematical formulation of these filters .
3、While the paper references temporal modules and DiT structures, it misses a deeper discussion on recent frequency-aware diffusion paradigms (e.g., FreeU), which also exploit the high/low-frequency dichotomy during the denoising trajectory.

---

> ### Author Rebuttal · Authors · 2026-03-26
>
> We sincerely thank the reviewer for raising these critical and insightful points, which have significantly enhanced the clarity and completeness of our manuscript.
>
> # W1: The fairness of the baselines regarding data curation.
> **Answer:** To clarify the baseline training setups: **both our method and UltraWan** utilize the **UltraVideo** dataset for training. **CineScale** employs a **private dataset** for their **LoRA fine-tuning**.
>
> To clarify the respective contributions of our data strategy and architectural design, we kindly refer to the ablation study in **Table 6 of the main paper**.  **Our model variant** evaluated without Data Selection and Augmentation (SA) ("Ours **w/o Data SA**") achieved an $FID_{patch}$ of **43.77**, a Realism score of **7.40**, and an Aesthetic Quality (AQ) score of **58.80**. These metrics still **substantially outperform** the corresponding scores for both **UltraWan** presented in **Tables 1 and 2 of the main paper**. This demonstrates that the significant performance **advantage of LUVE** fundamentally stems from our proposed **architectural design**. The Data SA strategies serve to push these already state-of-the-art baseline results even further. As shown in **Table 1** below, even without applying our Data SA strategies, our method maintains a definitive lead over the baselines.
>
> **Table 1. Comprehensive analysis for data augmentation.**
> | Model | Aesthetic Quality | $FID_{patch}$ | Realism | Detailness | Alignment |
> | :--- | :--- | :--- | :--- | :--- | :--- |
> | UltraWan-1K | 56.86 | 63.60 | 7.12 | 4.72 | 7.24 |
> | UltraWan-4K | 57.69 | 48.64 | 6.76 | 4.64 | 6.88 |
> | **Ours (w/o Data SA)** | **58.80** | **43.77** | **7.40** | **5.22** | **7.84** |
> | *Ours (Full)* | *59.78* | *41.03* | *7.64* | *5.36* | *7.90* |
>
> # W2: The mathematical formulation of LowPass(x) and HighPass(x) operations.
> **Answer:**  Let the input  of Low-Frequency Expert be denoted as $X$. We first transform the input into the frequency domain via a Fast Fourier Transform ($\mathcal{F}$) and apply a Gaussian-based low-pass filter $\mathcal{P}_G$ to extract the low-frequency components:
>
> $$
> X_{low} = LowPass(X)=\mathcal{F}^{-1}\Big(\mathcal{F}(X) \odot \mathcal{P}_G\Big),
> $$
>
> where $\odot$ denotes element-wise multiplication, and $\mathcal{F}^{-1}$ represents the  Inverse Fourier Transform. The Gaussian mask $\mathcal{P}_G$ is formulated as:
>
> $$
> \mathcal{P}_G(f) = \exp\left(-\frac{1}{2}\left(\frac{f}{d_c}\right)^2\right),
> $$
>
> Here, $f \in [-1, 1]$ represents the normalized frequency coordinates, and $d_c$ controls the bandwidth (cutoff frequency) of the filter.
>
> Similarly, for the High-Frequency Expert, the high-pass filter is defined as $\mathcal{P}_{H} = 1 - \mathcal{P}_G$, yielding:
>
> $$
> X_{high} = HighPass(X)= \mathcal{F}^{-1}\Big(\mathcal{F}(X) \odot (1 - \mathcal{P}_G)\Big).
> $$
>
>
> # W3: It misses a deeper discussion on recent frequency-aware diffusion paradigms (e.g., FreeU).
> **Answer:**  Recent studies have extensively explored the high/low-frequency dichotomy within diffusion denoising trajectories. FreeU [1] reveals an architectural frequency dichotomy in U-Net, demonstrating that the main backbone primarily governs low-frequency semantics, while skip connections introduce high-frequency details. It exploits this by re-weighting these components during inference. FRECAS [2] and FDG [3] further demonstrate that explicitly decoupling frequency components, whether through frequency-aware adapters or by decoupling Classifier-Free Guidance (CFG), can significantly mitigate the loss of high-frequency details and improve visual fidelity. Furthermore, STAR [4] and FreeLong [5] extend this dichotomy to the video domain. They observe that early diffusion steps establish low-frequency spatial structures and temporal dynamics, whereas later steps refine high-frequency details. Leveraging this property, they achieve video super-resolution and long video generation via loss constraints and training-free frequency blending, respectively. Following this fundamental insight, our LUVE is specifically designed for UHR video generation by fully leveraging this inherent coarse-to-fine frequency generation process. Unlike previous methods, LUVE explicitly trains Dual Frequency Experts to enhance the generative capabilities at distinct denoising stages. This explicitly trained component-level decoupling enables LUVE to not only synthesize authentic UHR details but also enhance the global semantic coherence and richness of the UHR videos.
>
> [1] FreeU: Free Lunch in Diffusion U-Net, CVPR 2024.
>
> [2] FreCaS: Efficient Higher-Resolution Image Generation via Frequency-aware Cascaded Sampling, ICLR 2025.
>
> [3] Guidance in the Frequency Domain Enables High-Fidelity Sampling at Low CFG Scales, Arxiv 2025.
>
> [4] STAR: Spatial-Temporal Augmentation with Text-to-Video Models for Real-World Video Super-Resolution, ICCV 2025.
>
> [5] FreeLong: Training-Free Long Video Generation with SpectralBlend Temporal Attention, NeurIPS 2024.

---

> > ### Author Rebuttal · Reviewer_Zhti · 2026-04-02
> >
> > The author's rebuttal addressed some of my concerns, but due to limited novelty, I will keep my score unchanged in the final decision.

---

> > > ### Author Response · Authors · 2026-04-03
> > >
> > > Thank you for acknowledging that our rebuttal addressed your technical concerns. We deeply appreciate your time and assure you that all clarifications will be incorporated into the revision. Regarding your remaining concern about novelty, we would like to take this final opportunity to respectfully but firmly clarify that LUVE is not merely an incremental architectural adjustment. Instead, it introduces **fundamental innovations** across the **problem setting, generation paradigm, architectural components, and theoretical insights**.
> > > # New Problem Setting:
> > > **Existing methods typically circumvent the fundamental challenges of UHR video generation through workarounds**. Training-free methods rely on T2V models never exposed to UHR data, leading to over-smoothed textures and a lack of inherent generative capacity for authentic UHR details. VSR-based approaches adopt a passive spatial upscaling pipeline that only enhances low-level textures, failing to generate meaningful semantic or structural details. **We are the first to systematically define and tackle the core challenges intrinsic to native UHR video generation**. We formally decouple these challenges into three distinct bottlenecks: **motion modeling, semantic planning, and detail synthesis**. LUVE is natively built to overcome these intrinsic barriers rather than bypassing them.
> > > # New Paradigm:
> > > Current cascaded models from both academia (e.g., FlashVideo, LaVie) and industry (e.g., Waver, Seedance, LongCat-Video) adhere to an "LR content generation + HR detail refinement (SR)" framework, where the HR stage acts merely as a passive perceptual enhancer unable to rectify semantic inaccuracies or synthesize rich content. To address the aforementioned challenges, LUVE pioneers a **new "LR motion generation + HR content completion" paradigm**. Departing from conventional frameworks, our LR stage exclusively provides motion priors, while the HR stage functions as an active content completer capable of rectifying LR semantic deviations and natively synthesizing authentic UHR video structures.
> > > # New Components:
> > > Rather than borrowing existing components, we propose entirely new components specifically tailored for UHR video generation:
> > >
> > > (1) Bypassing the computationally expensive VAE pixel-latent transitions, our VLUer, **the first trainable latent upsampler for cascaded UHR video generation**, operates entirely within the latent space. To mitigate the blocky artifacts and temporal inconsistencies that arise during the latent upsampling process, we introduce a lightweight decoder architecture guided by an RGB frame-difference loss. Our VLUer not only significantly reduces the computational cost and time overhead associated with standard RGB interpolation but also yields latent representations that more closely approximate true high-resolution video latents. This offers a novel perspective on performing upsampling directly within the latent space for cascaded HR video generation.
> > >
> > > (2) We introduce Dual Frequency Experts (DFE), **the first frequency decoupling mechanism specifically designed for UHR video generation**. This overcomes previous methods, achieving highly precise structural and detail synthesis. Our PSD analysis reveals that during UHR scaling, standard models struggle with drastic spatial expansion, causing low-frequency semantic inconsistencies and high-frequency detail degradation. To resolve this, the HFE injects authentic UHR details, which is proven by its ability to generate significantly stronger high-frequency PSD responses at low-noise stages (Fig. 2). The LFE actively corrects global semantics. We validate its efficacy through attention maps, which show coherent, object-specific activations rather than diffuse ones (Fig. 4). For specific details, please refer to our response to Reviewer 4KhT (W2 & Q1). For additional figures, please refer to the anonymous link: https://anonymous.4open.science/r/LUVE_Image-41D0/README.md.
> > > # New Insight:
> > > Beyond architectural designs, LUVE provides crucial new insights:
> > >
> > > (1) Scaling to UHR should not be limited to mere perceptual enhancement. **True UHR generation fundamentally advances semantic coherence, content fidelity, and visual richness rather than just improving sharpness**.
> > >
> > > (2) Scaling to UHR drastically expands spatial information, overwhelming standard DiTs during their natural low-to-high frequency denoising shift. **Dynamically reinforcing specific frequencies effectively mitigates both semantic inconsistencies and detail degradation**.

---

### Official Review · Reviewer_4KhT · 2026-03-09

**Soundness:** 3
**Presentation:** 4
**Significance:** 3
**Originality:** 2
**Overall Recommendation:** 4
**Confidence:** 4

**Summary:**

The authors investigate a pertinent problem: ultra-high-resolution video generation based on pretrained video diffusion backbones. The paper proposes LUVE, a three-stage cascaded framework consisting of low-resolution motion generation, latent-space video upsampling, and high-resolution content refinement with dual-frequency experts. Overall, the study's important result pertains to showing that this pipeline improves several quantitative metrics over recent UHR generation and VSR baselines under the reported setting, particularly against UltraWan and CineScale, which are also built on Wan2.1-1.3B.

The paper is generally easy to follow, and the low-frequency / high-frequency expert decomposition is the most interesting part of the method. However, I am not yet convinced that the work offers sufficient novelty or sufficiently strong evidence for the claimed mechanism.

**Compliance With Llm Reviewing Policy:**

Affirmed.

**Key Questions For Authors:**

Can the authors provide stronger evidence that the LF expert improves true semantic capability, rather than just helping convergence? It is difficult to see this clearly from the qualitative examples and quantitative results presented in the paper.

**Limitations:**

yes

**Strengths And Weaknesses:**

Strengths
1. The experimental results are broadly positive: LUVE outperforms UltraWan and CineScale on the reported VBench and UHR evaluation metrics, and also shows better perceptual scores than VSR baselines such as STAR and FlashVSR.
2. Among the proposed components, the dual-frequency expert design is the most interesting. The idea that low-frequency and high-frequency information may play different roles during denoising is at least intuitively plausible, and the ablation in Table 6 suggests that the full model is better than several reduced variants.
3. LUVE achieves somewhat better latency/memory than the two UHR generation baselines in their reported setup.

Weakness
1. The trainable latent upsampler is not a very new idea.
2. The paper’s explanation of why the LR/HR experts works is not fully convincing. In particular, although the LF/HF design is interesting, the claimed semantic improvement from the LF expert is not clearly supported by the metrics or qualitative examples. I am not convinced that the proposed expert trick really addresses the core challenges raised in the introduction for video super-resolution / UHR video generation. This feels more like a trick for accelerating convergence. It would be better to provide a more rigorous mathematical analysis and broader experimental validation.
3. The overall contribution feels incremental rather than fundamentally novel.

---

> ### Author Rebuttal · Authors · 2026-03-29
>
> # W1:
> **Answer:** We appreciate the valuable comments. To the best of our knowledge, we are the **first** to introduce a trainable latent upsampler for latent-cascaded UHR video generation. To mitigate the blocky artifacts and temporal inconsistencies that arise during the latent upsampling process, we introduce a lightweight decoder architecture guided by an RGB frame-difference loss. As demonstrated in Tables 5 and 12 of the main paper, our VLUer not only significantly reduces the computational cost and time overhead associated with standard RGB interpolation but also **yields latent representations that more closely approximate true high-resolution video latents**. This offers a novel perspective on performing upsampling directly within the latent space for cascaded HR video generation.
> # W2 & Q1:
> **Answer:** To investigate the frequency dynamics of video diffusion models during denoising, we perform Power Spectral Density (PSD) analysis on Wan2.1. As **Fig. 1** reveals, the model exhibits frequency-progressive behavior: primarily capturing low-frequency components at high-noise stages, then shifting to high-frequency details at low-noise stages. However, UHR scaling drastically expands spatial information, causing low-frequency semantic inconsistencies and high-frequency detail degradation.
> To examine this, we compare PSD distributions among our full model, the baseline (w/o Experts), and the ground truth (GT) (**Figs. 2, 3**). For fair comparison, models generate videos based on Gemini 3 captions derived from real-world UHR videos (GT). Results show that at low-noise stages, our method yields significantly stronger, GT-aligned high-frequency responses than the baseline, proving the HF expert effectively enhances UHR detail reconstruction.
> While our method also achieves GT-aligned PSD at high-noise stages, PSD reflects spectral magnitude rather than semantic coherence. Thus, attention map analysis (**Fig. 4**) confirms the LFE's role: our method generates coherent, object-specific attention, whereas discarding the LFE causes diffuse, non-specific activations. Additional visual comparisons (**Fig. 5**) further demonstrate the LFE fundamentally improves true semantic capabilities. To quantitatively evaluate this, we designed two metrics: $FID_{lowfre}$ (which measures the FID of low-frequency components) and *Semantic-Plausibility* (which assesses the rationality and quality of semantic content using an MLLM). As reported in **Table 1**, our full model achieves significant improvements across both metrics. Together, these qualitative and quantitative results validate that the LFE effectively mitigates the challenges of capturing and modeling global semantic layouts. **For additional figures, please refer to the anonymous link**: https://anonymous.4open.science/r/LUVE_Image-41D0/README.md.
>
> **Table 1. Ablation the LFE on semantic modeling.**
> | Model | $FID_{lowfre}$ $\downarrow$ | *Semantic-Plausibility* $\uparrow$ |
> |:-|:-|:-|
> |UltraWan-1K|104.65|8.48|
> |UltraWan-4K|109.03|7.60|
> |Ours (w/o LFE)| 103.00|8.40|
> |**Ours (Full)**|**97.26**|**8.88**|
> # W3:
> **Answer:** While LUVE integrates existing concepts, the way we structure the cascade is fundamentally novel. Compared to current cascaded HR video gen models from academia (e.g., FlashVideo, LaVie) and industry (e.g., Waver, Seedance), LUVE exhibits fundamental innovation across its generation paradigm, components, and theoretical insights.
> ### **New Paradigm**:
> Current cascaded models adhere to an "LR content generation + HR detail refinement" pipeline, where the HR stage acts merely as a passive perceptual enhancer unable to rectify semantic inaccuracies or synthesize rich content. In contrast, LUVE pioneers a **new "LR motion generation + HR content refinement" paradigm**. Breaking free from passive sharpening, our HR stage functions as an active content completer, rectifying LR semantic deviations and natively synthesizing authentic UHR video.
> ### **New Component**:
> (1) Bypassing the computationally expensive VAE pixel-latent transitions, our VLUer, the first trainable video latent upsampler, operates entirely within the latent space, significantly reducing computational overhead.
> (2) Our DFE dynamically decouple generation: the LFE corrects the global semantic while the HFE injects UHR textures. This overcomes previous methods, achieving highly precise structural and detail synthesis.
> ### **New Insight**:
> Beyond architectural designs, LUVE provides crucial new insights: (1) Scaling to UHR should not be limited to mere perceptual enhancement. True UHR generation fundamentally advances semantic coherence, content fidelity, and visual richness rather than just improving sharpness. (2) Scaling to UHR drastically expands spatial information, overwhelming standard DiTs during their natural low-to-high frequency denoising shift. By dynamically reinforcing specific frequencies, our method mitigates both semantic inconsistencies and detail degradation.

---

> > ### Author Rebuttal · Reviewer_4KhT · 2026-04-01
> >
> > My concerns have been adequately addressed.

---

> > > ### Author Response · Authors · 2026-04-03
> > >
> > > We deeply appreciate the time and expertise you have invested in reviewing our manuscript. We are pleased to know that our previous responses have successfully resolved your queries. Your insightful suggestions have been instrumental in enhancing the clarity and overall quality of our work. Furthermore, we assure you that all supplementary materials and clarifications will be carefully incorporated into the revised manuscript.

---

### Official Review · Reviewer_c2GU · 2026-03-09

**Soundness:** 3
**Presentation:** 2
**Significance:** 2
**Originality:** 2
**Overall Recommendation:** 4
**Confidence:** 3

**Summary:**

This paper aims to address the challenge of generating ultra-high-resolution videos with diffusion models. The authors propose a new module, Video Latent Upsampling (VLU), to mitigate feature distortion and manifold deviation during the upsampling process. To further improve video quality, the method introduces low-frequency and high-frequency experts, designed to balance the trade-off between semantic consistency and fine-grained detail generation. The experimental results, along with ablation studies, demonstrate the effectiveness of the proposed components and show that the method achieves state-of-the-art performance.

**Compliance With Llm Reviewing Policy:**

Affirmed.

**Final Justification:**

I would like to increase my score to WA. From the new table 2 in rebuttal, the LoRA expert setting does not appear to work, which suggests that the high-pass and low-pass modules are indeed effective. I also encourage the authors to provide more details about Table 6, specifically clarifying how the LoRA expert setup differs from your proposed method, in order to better highlight the contribution. I strongly believe this design deserves more space in the paper, particularly with a clearer and more thorough ablation study of this module.

**Key Questions For Authors:**

1. Could the authors clarify whether the baseline model was fine-tuned on the same dataset?

2. Does the proposed pipeline generalize to video super-resolution? If so, how does it compare with other baselines in terms of PSNR?

**Limitations:**

Yes

**Strengths And Weaknesses:**

Strengths

1. This paper introduces a technically sound paradigm. The proposed improvements appear promising, as supported by the experimental results and ablation studies.
2. Training VideoINR as a video latent upsampler is an interesting design choice and provides a novel perspective for video generation.
3. The proposed method achieves state-of-the-art performance on the evaluated benchmarks.


Weakness:

1. The overall novelty of this paper seems limited. The main contribution appears to be the Video Latent Upsampling (VLU) module. However, using an additional network to refine decoder features or latent representations has been widely explored in the super-resolution literature [1,2,3]. The paper does not clearly explain how this work differs from those approaches.

2. The motivation behind the Low-Frequency Expert (LFE) and High-Frequency Expert (HFE) modules is not sufficiently convincing. The observation that global semantic structures are reconstructed during the high-noise stage while fine-grained details emerge in the low-noise stage has already been discussed in prior work. The paper lacks deeper analysis explaining why these experts are necessary and what limitations of previous methods they address. More discussion or empirical evidence is needed to clarify why this design works better than existing approaches.

3. Some qualitative results raise concerns about the reliability of the proposed pipeline. In the supplementary material (the final section comparing with Base, VEhancer, and STAR), the structure of objects such as trees changes significantly, while other methods preserve the original structure. This suggests that the VLU module may modify regions that were already reasonable in the base image, rather than only correcting problematic areas. As a result, the method may sacrifice consistency with the original input, potentially reducing the reliability of the generated results. This also raises the concern that the observed improvements may mainly come from fine-tuning or data augmentation.

4. Overall, the paper appears somewhat incremental, as many components are derived from existing works. The paper would benefit from a clearer explanation of what new insights or capabilities emerge from combining these components, beyond incremental improvements in performance.



References

1. Wang, J., Yue, Z., Zhou, S., Chan, K. C., & Loy, C. C. (2024). Exploiting diffusion prior for real-world image super-resolution. International Journal of Computer Vision, 132(12), 5929-5949.
2. Zhou, S., Yang, P., Wang, J., Luo, Y., & Loy, C. C. (2024). Upscale-a-video: Temporal-consistent diffusion model for real-world video super-resolution. In Proceedings of the IEEE/CVF conference on computer vision and pattern recognition (pp. 2535-2545).
3. Yang, X., He, C., Ma, J., & Zhang, L. (2024, September). Motion-guided latent diffusion for temporally consistent real-world video super-resolution. In European conference on computer vision (pp. 224-242). Cham: Springer Nature Switzerland.

---

> ### Author Rebuttal · Authors · 2026-03-29
>
> # W1 & W4:
> **Answer:**  We thank the reviewer for the constructive comments. Unlike the recommended Super-Resolution (SR) works [1, 2, 3], LUVE exhibits **fundamental differences** in problem setting, generation paradigm, and component innovation:
> ### **Problem Setting:**
> SR models primarily address adaptation issues (e.g., perception-distortion tradeoff, temporal consistency) when transferring pre-trained diffusion models to SR tasks. Conversely, LUVE targets core challenges intrinsic to native UHR generation, overcoming inherent bottlenecks like motion modeling, semantic inconsistencies, and detail synthesis (Fig. 2 of paper).
> ### **New Paradigm:**
> Existing methods follow a "LR content generation + SR detail sharpening" post-processing paradigm. Here, SR acts merely as a perceptual enhancer, struggling with UHR semantic integrity. Conversely, LUVE introduces a **new cascaded paradigm: "LR motion generation + HR content refinement"**. Our HR stage acts as an active content completer, rectifying LR semantic deviations and natively synthesizing rich UHR content.
> ### **Component Innovation:**
> (1) Unlike existing SR models relying on VAE pixel-latent transitions, our VLUer is a lightweight, intra-latent operator tailored for cascaded generation. As the **first** trainable video latent upsampler, VLUer bypasses VAE encoding/decoding bottlenecks, significantly reducing computational overhead.
> (2) Our Dual-Frequency Experts dynamically decouple generation: the LFE corrects the global semantic layout, while the HFE injects UHR textures. This mechanism overcomes SR limitations, achieving precise structural and detail generation.
> ### **New insight :**
> Beyond architectural designs, LUVE provides crucial new insights: (1) Scaling to UHR should not be limited to mere perceptual enhancement. True UHR generation fundamentally advances semantic coherence, content fidelity, and visual richness rather than just improving sharpness. (2)  Scaling to UHR drastically expands spatial information, overwhelming standard DiTs during their natural low-to-high frequency denoising shift. By dynamically reinforcing specific frequencies, our method mitigates both semantic inconsistencies and detail degradation (see PSD analysis below).
> # W2 :
> **Answer:** To investigate the frequency dynamics of video diffusion models during denoising, we perform Power Spectral Density (PSD) analysis on Wan2.1. As shown in **Fig. 1**, the model shows a clear frequency behavior: primarily capturing low-frequency components in the high-noise stage, then gradually shifting focus toward high-frequency details in the low-noise stage. However, scaling to UHR substantially expands spatial information, resulting in low-frequency semantic structural inconsistencies and high-frequency detail degradation. To examine this, we compare the PSD distributions of our full model, the baseline (w/o Experts), and the ground truth (GT) (**Figs. 2 and 3**). For a fair comparison, we use real-world UHR videos as the Ground Truth (GT). The evaluated models then generate videos using prompts derived from these GT videos via Gemini 3. Results demonstrate that in the low-noise stage, our method produces significantly stronger high-frequency responses than the baseline, aligning closer with the GT. This indicates the HFE effectively enhances UHR detail reconstruction. Moreover, although our method also achieves GT-aligned PSD distributions in the high-noise stage, PSD primarily reflects spectral magnitude rather than semantic coherence. Thus, we further analyze attention maps to understand the LFE's role in modeling global semantic structures. As shown in **Fig. 4**, ours generates object-specific and coherent attention maps, whereas discarding the LFE results in diffuse and non-specific activations. This validates that the LFE effectively resolves the challenge of capturing and modeling the global semantic layout. **For additional figures, please refer to the anonymous link**: https://anonymous.4open.science/r/LUVE_Image-41D0/README.md.
> # W3 :
> **Answer:** As clarified above, LUVE operates as a generation paradigm rather than SR framework. The LR stage solely provides a motion prior; thus, the HR stage does not aim to maintain strict consistency with the LR input, but to actively refine structure and content. The reviewer’s observation that " trees change significantly" perfectly illustrates our intended behavior. As seen in Fig. 16 of paper,  our method generates highly realistic trees (e.g., distinct branches). We have thoroughly validated the superiority of this paradigm through extensive qualitative and quantitative experiments. Regarding data augmentation, please refer to our response to Reviewer Zhti (W1).
> # Q1 :
> Our LUVE and UltraWan utilize the UltraVideo dataset. CineScale employs a private dataset.
> # Q2:
> LUVE is designed as a generative paradigm rather than a degradation-restoration model. Thus, it is challenging to directly generalize it to conventional VSR tasks.

---

> > ### Author Rebuttal · Reviewer_c2GU · 2026-04-03
> >
> > I am not convinced by the authors’ assumption that trainable video latent upsampler in video generation is a novel topic, as this has been explored VSR tasks (upsampler 3D convolution). In my view, the main contribution is the LR/HR expert design; however, the rebuttal does not sufficiently justify why this design is necessary rather than a heuristic trick. The reported gains may also partly come from larger training data or increased parameter capacity. In particular, it would be important to show whether simply adding LoRA attention/FFN modules without LR/HR separation yields similar performance. Moreover, the observation that diffusion models progressively capture global semantics in early denoising stages and refine fine-grained details in later stages has already been well established in prior work. Finally, the method seems inherently tied to a multi-step diffusion pipeline and may not generalize to one-step diffusion settings.
> >
> > Overall, I still tend to keep my score as WR.

---

> > > ### Author Response · Authors · 2026-04-04
> > >
> > > We sincerely thank you for your deep and insightful questions, which are highly valuable in enhancing the quality of our paper. We also appreciate your acknowledgment that some of the issues raised in the previous round have been successfully resolved.
> > >
> > > Below, we address your new concerns point by point:
> > >
> > > ## 1. Regarding the video latent upsampler:
> > > We acknowledge that INR and 3D convolutions have been extensively explored in VSR tasks. However, our proposed video latent upsampler is fundamentally different, as it is exclusively tailored for cascaded UHR video generation. We design the video latent upsampler to address key limitations in existing HR video cascaded pipelines. Specifically, direct latent interpolation typically leads to feature distortion and manifold deviation. Conversely, performing interpolation in the RGB space imposes heavy memory and latency burdens due to the VAE codecs. Our experiments (Table 5 of the paper) demonstrate that our proposed video latent upsampler effectively resolves these issues, achieving the best results while maintaining high inference speed.
> > >
> > > ## 2. Regarding the LR/HR experts:
> > > We understand your concern that the performance improvements might stem from larger training data or increased parameter capacity rather than the proposed design.
> > > ### (1) Training Data:
> > > We strictly used **the same training data and pre-trained model** as UltraWan. As shown in **Table 1**, which was already reported in our original manuscript, even without any data augmentation, our model achieves substantial performance improvements over UltraWan. This clearly demonstrates that the performance gains are not simply driven by the training data.
> > >
> > > **Table 1. Comprehensive analysis for training data.**
> > > | Model | Aesthetic Quality | $FID_{patch}$ | Realism | Detailness | Alignment |
> > > | :--- | :--- | :--- | :--- | :--- | :--- |
> > > | UltraWan-1K | 56.86 | 63.60 | 7.12 | 4.72 | 7.24 |
> > > | UltraWan-4K | 57.69 | 48.64 | 6.76 | 4.64 | 6.88 |
> > > | Ours (w/o Data SA) | 58.80 | 43.77 | 7.40 | 5.22 | 7.84 |
> > > | Ours | 59.78 | 41.03 | 7.64 | 5.36 | 7.90 |
> > >
> > > ### (2) Further Analysis For LR/HR Experts:
> > > To further analyze our LR/HR Experts, **Table 2** presents an ablation study comparing them against pure LoRA Experts (without LR/HR separation). Note that a corresponding quantitative analysis was **already** included in **Table 6 of our original manuscript**. The **LoRA Experts** share the exact same experimental setup as our Frequency Experts: they are applied to the attention modules during the high-noise stage and the FFN modules during the low-noise stage, with an identical LoRA rank of 96. As shown in our results, **while using pure LoRA Experts brings minor improvements, there remains a massive performance gap compared to our proposed Frequency Experts**. Additionally, we provide supplementary Power Spectral Density (PSD) analysis in **Figures 6 and 7** (Anonymous link: https://anonymous.4open.science/r/LUVE_Image-41D0/README.md). **These visualizations clearly show that pure LoRA Experts fail to help the model capture the distinct frequency components corresponding to different denoising stages**. This strongly validates the necessity of our targeted LR/HR expert design and proves it is not merely a heuristic trick of increasing parameters. We agree that the progressive nature of generation has been well-documented in existing literature. However, unlike previous works, we analyze and leverage this process strictly through the lens of the frequency domain, resolving the problem by selectively enhancing specific frequencies during training.
> > >
> > > **Table 2. Ablation with pure LoRA Experts.**
> > > | Model | Aesthetic Quality | $FID_{patch}$ | Realism | Detailness | Alignment |
> > > | :--- | :--- | :--- | :--- | :--- | :--- |
> > > | w/o Experts | 58.57 | 46.48 | 7.00 | 4.92 | 7.44 |
> > > | LoRA Experts  | 58.65 | 47.03 | 7.28 | 4.90 | 7.48 |
> > > | Ours | 59.78 | 41.03 | 7.64 | 5.36 | 7.90 |
> > >
> > > ## 3. Regarding the generalization to one-step diffusion settings:
> > > We agree with your insightful observation. Our current framework is indeed challenging to adapt to a strict one-step diffusion setting, which currently limits its inference efficiency. However, **generalizing the HR stage to a two-step diffusion setting is highly promising** (i.e., allocating one step for the Low-Frequency expert and one step for the High-Frequency expert). This adaptation would significantly enhance inference efficiency while preserving the benefits of our frequency-aware design. We are actively exploring this direction and consider it a key focus for our future work.
> > >
> > > **We hope these clarifications and the additional analyses address your concerns. Thank you again for your constructive feedback.**

---

### Decision · Program_Chairs · 2026-04-30

**Decision:**

Accept (regular)

**Comment:**

This paper was reviewed by 4 experts in the field. After discussion, the reviewers still hold a mixed review to this work. The rating is 4(weak accept), 4(weak accept), 4(weak accept), 3(weak reject).

Overall, reviewers agree that this is a high-quality work, as achieves SOTA 4K video generation, but only uses an efficient 1.3B base model. Also, the proposed dual-frequency design is simple and effective and the latent upsampler is also interesting.

Still, reviewers raised several concerns to this work. The concern includes  a potentail incremental novelty compared to standard cascaded pipelines, and running speed (as it is a multi-step diffusion).

Based on this, the decision of this work is to Accept. Still, we strongly recommend the authors carefully read all reviewers’ final feedback and revise the manuscript as suggested in the final camera-ready version if being accepted.